# CRAFT: Concept Recursive Activation FacTorization for Explainability

## Abstract

Despite their considerable potential, concept-based explainability methods have received relatively little attention, and explaining *what*'s driving models' decisions and *where* it's located in the input is still an open problem. To tackle this, we revisit unsupervised concept extraction techniques for explaining the decisions of deep neural networks and present CRAFT – a framework to generate concept-based explanations for understanding individual predictions and the model's high-level logic for whole classes. CRAFT takes advantage of a novel method for recursively decomposing higher-level concepts into more elementary ones, combined with a novel approach for better estimating the importance of identified concepts with Sobol indices. Furthermore, we show how implicit differentiation can be used to generate concept-wise attribution explanations for individual images. We further demonstrate through fidelity metrics that our proposed concept importance estimation technique is more faithful to the model than previous methods, and, through human psychophysic experiments, we confirm that our recursive decomposition can generate meaningful and accurate concepts. Finally, we illustrate CRAFT's potential to enable the understanding of predictions of trained models on multiple use-cases by producing meaningful concept-based explanations.[*]

## 1   Introduction

Interpreting the decisions of modern machine learning models such as neural networks remains a major challenge. The need for robust and reliable explainability methods has never been more urgent as machine learning is being applied to an ever increasing range of domains, including safety critical ones. The application of the General Data Protection Regulation law (GDPR) [1] in the European Union has drawn the attention of the general public to the rights they should have on their data. This kickstarted a race for other needs, with more and more regulation agencies asking for the right for AI decisions to be explainable to users – e.g. European AI act [2], EASA concepts for design assurance [3].

In order to try to meet this need, an array of explainability methods have already been proposed. Most of these methods aim at explaining what inputs (or pixels in an image) are driving the model's decision. These so-called attribution methods yield heatmaps that indicate the importance of individual pixels. Among the most notable ones is LIME [4], which was initially developed to try to locally – that is, at an instance level – understand models' predictions to identify possible biases in vision models. Multiple improvements have since been introduced – either by better harnessing the information provided by gradients to estimate the importance of individual pixels [5, 6, 7, 8, 9, 10, 11, 12], leveraging image perturbations to evaluate the sensitivity of a model's output [13, 14] or, more recently, via the use of formal methods to generate explanations [15].

---

[*]Our code is available at `anonymous.4open.science/r/craft-concept-explanation-4351`.

However, all the aforementioned methods focus on one side of explainability – answering the question of *where* – i.e., where in an image are the pixels that are critical to the decision located. They leave the question of *what* – i.e., what visual features are actually driving decisions – entirely open. We argue that this limitation is one of the main reasons why these methods fail in some cases to help users, for instance, identify the source of a system's bias or its failure cases as shown in [16]. Feature visualization methods [17, 18] characterize the selectivity of individual neurons (or neural channels or arbitrary directions in the neural activation space) via the synthesis of input stimuli which maximize their responses and can partially answer this question. Still in this vein, [19, 20, 21] proposed to use the training dataset to identify the samples that contribute the most to the model's decision. Finally, closer to our work, a new line of research has recently been initiated [22] based on high-level concepts. The goal of this branch is to find humanly interpretable concepts in the activation space of a layer in a neural network. This approach can give positive results, but in its original formulation, it requires prior knowledge on the relevant concepts, and more importantly, the labeling of a dataset for each of the concepts we want to extract. Hence, several works have proposed to automate the concept search based only on the training dataset and without explicit human supervision. The most prominent of these techniques, ACE [23], uses a combination of segmentation and clustering techniques, but requires heuristics to remove outliers. This method unlocks the possibility of large scale concept extraction without additional labeling or human supervision. Nevertheless, it suffers from several problems: each segment can only belong to one cluster, the choice of the layer from which to retrieve the concepts is not clear, and the amount of information lost during the outlier rejection phase can be a cause of concern. More recently, [24] proposes to leverage matrix decompositions on internal feature maps to discover concepts.

It is important to note that current work does not offer a link between their global and local explanations, nor do they offer an answer to the question of which layer to choose to perform the decomposition. Building up on these conclusions, we revisit these concept extraction techniques by using Non-Negative Matrix Factorization (NMF) and propose 3 different ingredients to answer these questions simultaneously, thereby introducing CRAFT, a new automatic concept extraction method.

We can summarize our main contributions as follows:

- A novel approach for the automated extraction of high-level concepts learned by deep neural networks.
- A recursive procedure to automatically decompose concepts into sub-concepts, starting with the last layer of the model and working our way inwards. We validate the benefit of this recursivity – i.e. decomposing concepts into sub-concepts – with human psychophysic experiments which show that (i) that the decomposition of a concept yields more coherent sub-concepts (ii) the groups of points formed by these sub-concepts are more refined and appear meaningful to humans (expert or non-expert).
- A novel technique to quantify the importance of individual concepts on a model's predictions using Sobol indices coming from the field of Sensitivity Analysis.
- A novel *Concept Attribution Map* (CAM) method to backpropagate each of the concept values independently into the pixel space by leveraging the implicit function theorem, allowing us to locate the concept in a given input image. This effectively unlocks the ability to apply all the white-box [5, 6, 7, 8, 9, 12, 25] and black-box [4, 26, 13, 14] explainability techniques in the literature to obtain concept-wise attribution maps.
- A demonstration of the approach combining local and global explanations to accurately explain predictions and understand complex failure cases.

## 2   Related Work

**Explaining *where***   The widespread use of black-box machine learning methods including deep convolutional neural networks in myriads of computer vision tasks prompted a need to understand where in the input image the model looked to make predictions. These explanatory heatmaps can be generated through completely different approaches depending on whether access to gradients is provided. If it is indeed the case, there's a plethora of different methods that harnesses intermediary information inside the neural network to create these explanations [5, 8, 7, 28, 6, 29, 9, 12]. However, they have been found to induce confirmation bias [30] and to be vulnerable to adversarial attacks [31]. Somewhat differently, there are other methods [10, 11] that harness gradients to optimize masks to maximize the impact on the predictions, and thus determine the most important parts of the input for the model. However, if only the input and its corresponding output are available, other

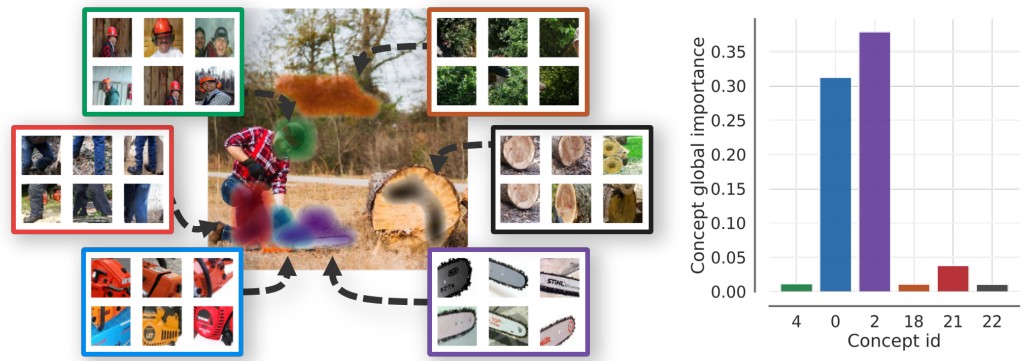

Figure 1: **CRAFT Results for the prediction 'Chain Saw'.** First, our method uses NMF to extract from the train set (ILSVRC2012 [27]) the most relevant concepts used by the network (ResNet50V2). Then, the global influence of these concepts on predictions is measured using Sobol indices (right panel). Finally, the method provides local explanations through *Concept Attribution Maps* (heatmap associated to a concept, and computed using grad-CAM by backpropagating through the NMF concept values with implicit differentiation). Besides, concepts can be interpreted by looking at crops that maximize the NMF coefficients. For the class 'Chain Saw', the detected concepts seem to be: $\mathcal{C}_0$ for the chainsaw engine, $\mathcal{C}_2$ for the saw blade, $\mathcal{C}_4$ for the human head, $\mathcal{C}_{18}$ for the vegetation, $\mathcal{C}_{21}$ for the jeans and $\mathcal{C}_{22}$ for the tree trunk.

techniques exist that enable the generation of attribution maps by locally estimating the importance of each input pixel: LIME [4], RISE [13], and more recently, an attribution method based on Sobol indices [14, 32]. Crucially, they propose to input perturbed versions of the example one wishes to explain and either construct a linear model to determine the importance of each region of the input, leverage Monte-Carlo methods to this end, or compute the Sobol indices [32] associated to them as a measure of their influence on the model. Concretely, we will be exploiting all this literature to locate the important parts of images with respect to what we will call "high-level" concepts by generating concept-wise attribution maps.

**Explaining «what»** There have been studies [33, 34] that indicate that CNNs trained on the ImageNet dataset [27] rely heavily on textures to classify, and largely disregard the shapes. For this reason, some researchers suggest that attribution maps might not be enough to explain models' predictions [17], and that explainability methods revealing the role of the textures are a must. Namely, in [18] and [17], explanations are generated as the inputs that would maximize the neural activation of a given layer with respect to a given class. However, these explanations may not be easily interpretable by humans. Finally, other approaches suggest to modify the structure of the neural network, either by constraining the convolutional layers to naturally provide visual explanations [35], or by forcing it to generate prototypes for the classes [36], but our main focus are post-hoc methods that can be applied to pre-trained neural networks and don't need further training.

**Concept discovery.** In [22], Kim et al. proposed an alternative to explaining the *what*: they built a database with different concepts (such as "stripes") to extract a concept vector in the latent space of a given layer. Then they proposed to estimate the importance of this concept vector using the directional derivative of the model's predictions with respect to this concept vector. However, it is a supervised approach, and thus, only applicable when we have prior knowledge of the concepts in play. The natural extension of this idea is automatic discovery of concepts in an unsupervised fashion, without the need for prior knowledge or labelled concept datasets. As such, in [23], a technique is proposed to discover these "high-level" concepts: they perform segmentation at different resolutions on patches of images, cluster them and select the most significant based on perception and Testing with Concept Activation Vectors (TCAV) [22] scores. However, the quality of the result is highly dependent on the segmentation scheme and on the layer used for perception scores. Building up on this technique, [24] propose to generate a bank of concepts for each class by performing dimensionality reductions on the activation maps flattened over the channel dimension. Once the factorization done, the reconstruction of the activation of the image can then be interpreted as a combination of a set of concepts and a coefficient associated to these concepts. Not all factorization-based methods are equal though. Their large-scale human experiments show an interesting trend: Non-negative Matrix Factorization (NMF) is widely preferred over Principal Component Analysis (PCA) or ACE for generating meaningful

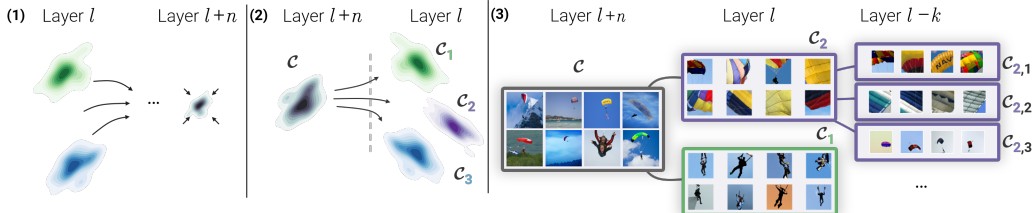

Figure 2: **(1) Neural Collapse (Amalgamation)** classifiers need to be able to linearly separate each class at the last layer, and to do this, the activations of the same class must merge during the forward pass until they all converge to the one-hot vector of the class in the logits layer. This may result in activations that are too concentrated to be broken down into meaningful concepts. **(2) Recursive process** When a concept is not understood (e.g., $\mathcal{C}$), we propose to decompose it into multiple sub-concepts (e.g., $\mathcal{C}_1, \mathcal{C}_2, \mathcal{C}_3$) using the activations from an earlier layer to overcome the aforementioned neural collapse issue. **(3) Example of concept recursive decomposition** using CRAFT on the class 'Parachute' of ILSVRC2012 [27].

concepts for humans. Finally, [37] defines the notion of *completeness* of a concept bank and proposes a method to learn a complete set of concepts using Shapley values [26].

## 3  Overview of the method

In this section, we first describe our Concept Activations Factorization method by pointing out the differences that set our technique apart from previous work. We then proceed to introduce the three new ingredients that make up CRAFT: (1) a method to recursively decompose concepts into sub-concepts, (2) a new approach to better estimate the importance of extracted concepts and (3) how we unlock any attribution method to create *Concept Attribution Maps*, using implicit differentiation [38, 39, 40].

**Notation**    In this work, we consider a general supervised learning setting, where $(\boldsymbol{x}_1, ..., \boldsymbol{x}_n) \in \mathcal{X}^n$ are $n$ points and $(y_1, ..., y_n) \in \mathcal{Y}^n$ their associated labels. Unless specified, all points are assumed to have the same labels. We are given a (machine-learnt) black-box predictor $\boldsymbol{f} : \mathcal{X} \to \mathcal{Y}$, which at some test input $\boldsymbol{x}$ predicts the output $\boldsymbol{f}(\boldsymbol{x})$. Without loss of generality, we assume that $\boldsymbol{f}$ is a neural network composed of $k$ layers, and we denote $\boldsymbol{f}(\boldsymbol{x}) = \boldsymbol{h}_k \circ \boldsymbol{h}_{k-1} \circ ... \circ \boldsymbol{h}_1(\boldsymbol{x})$ with $\boldsymbol{h}_l(\boldsymbol{x}) \subseteq \mathbb{R}^p$ being the intermediate activations for the layer $l$ and $\boldsymbol{h}_l(\boldsymbol{x})_i$ an activation for the same layer. Further, we require non-negative activations: $\boldsymbol{h}_l(\boldsymbol{x})_i \geq 0 : \forall i \in \{1, ..., p\}$, which amounts to choosing a layer whose activation function $\boldsymbol{\sigma}_l(\boldsymbol{x}) \geq 0$. In particular, this assumption is verified by any architecture that utilizes *ReLU*, but any non-negative activation function works. Finally, we denote $\boldsymbol{h}_{l,k}$ the function going from the layer $l$ to the output of the model $\boldsymbol{f}$.

### 3.1  Concept Activations Factorization

As illustrated in Fig.3, we propose to use Non-negative matrix factorization activations to find a basis of concepts. Inspired by ACE [23], we will use sub-regions of images to attempt to identify coherent concepts. Instead of using segmentation – which naturally introduces artifacts due to the inpainting required by a baseline value –, we start by taking random crops of each image in our dataset (e.g, a set of points that the model predicts as belonging to the same class) to form an auxilary dataset $\boldsymbol{X} \in \mathbb{R}^{n \times d}$ such that $\boldsymbol{X}_i = \boldsymbol{\tau}(\boldsymbol{x}_i)$ with $\boldsymbol{\tau}$ a crop function. Given a layer $l$, we obtain the activations for the random crops $\boldsymbol{A} = \boldsymbol{h}_l(\boldsymbol{X}) \in \mathbb{R}^{n \times p}$. In the case where $\boldsymbol{f}$ is a convolutional neural network, a global average pooling is applied on the activations. We recall that all the elements of $\boldsymbol{A}$ are non-negative real numbers.

We are now ready to apply Non Negative Matrix Factorization (NMF) to decompose the positive activations $\boldsymbol{A}$, into a product of non-negative, low rank matrices $\boldsymbol{U}(\boldsymbol{A}) \in \mathbb{R}^{n \times r}$ and $\boldsymbol{W} \in \mathbb{R}^{p \times r}$, with:

$$\min_{\boldsymbol{U} \geq 0, \boldsymbol{W} \geq 0} \frac{1}{2} \|\boldsymbol{A} - \boldsymbol{U}\boldsymbol{W}^T\|_F^2 \qquad (1)$$

Where $|| \cdot ||_F$ denotes the Frobenius norm. One of the appealing properties of NMF is the low rank constraint $r \ll \min(n, p)$. Simply put, NMF can be understood as the joint learning of $\boldsymbol{W}$, a dictionary of CAVs – "concept bank" in Figure 3 – that maps a $\mathbb{R}^p$ basis onto $\mathbb{R}^r$, and $\boldsymbol{U}$ the coefficients of vectors $\boldsymbol{A}$ expressed in this new basis. The minimization of the reconstruction error

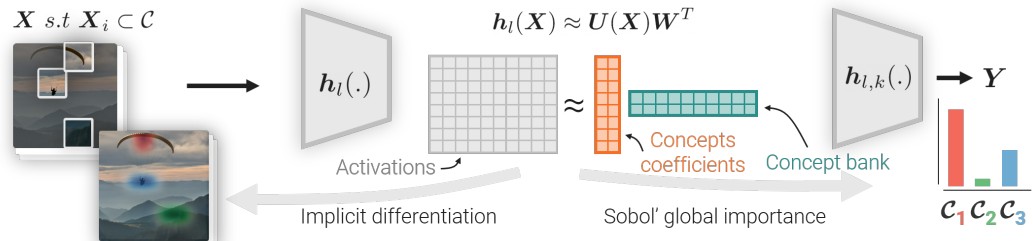

Figure 3: **Overview of CRAFT.** Starting from a set of crops $X$ containing a concept $\mathcal{C}$ (e.g., crops images of the class Parachute), we send random crops to a layer $l$ to get activations it $h_l(X)$. We then factorize the activation into two lower rank matrices, $U$ and $W$. $W$ is what we call a concept bank (a base of concepts), while $U$ corresponds to the coefficients in this new basis. We then extend the method with 3 new ingredients: (1) the recursivity by proposing to re-decompose a concept (e.g., take a new set of point containing $\mathcal{C}_1$) at an earlier layer $l' < l$, (2) a better importance estimation using Sobol indices and (3) leveraging implicit differentiation to generate *Concept Attribution Maps* allowing to localize concepts in an image.

$\frac{1}{2}\|A - UW\|_F^2$ ensures that the new basis contains (mostly) relevant concepts. Intuitively, the non-negativity constraints $U \geq 0, W \geq 0$ encourage (i) the sparsity of $W$ (useful for creating disentangled concepts), (ii) the sparsity of $U$ (convenient for selecting a minimal set of useful concepts) and (iii) the imputation of missing data [41], which corresponds to the sparsity pattern of *post-ReLU* activations $A$. We shall also note that each original activation $A_i$ coming from the input $x_i$ can be approximated by its reconstruction $h_l(\tau(x_i)) = U_i W^T = \sum_{j=1}^{r} U_{i,j} W_j^T$. This approach is attractive as each activation can be understood as a composition of concepts.

While other methods in the literature solve a similar problem (such as low rank factorization using SVD or ICA), the NMF has stepped up as both fast, effective and is known to yield meaningful concepts to humans [42, 43, 24]. Finally, once the concept bank $W$ is precomputed, we can associate the concept coefficients $U(x)$ to any new input $x$ (e.g a full image) by solving the underlying Non-Negative Least Squares (NNLS) problem $\min_{U \geq 0} \frac{1}{2}\|h_l(x) - U(x)W^T\|_F^2$, and therefore have its decomposition in the concept base.

In essence, the core of our method can be summarized as follows: using a set of images, we re-interpret their embedding at a given layer $l$ as a composition of concepts that can be easily understood by humans. In the next section we show how we can recursively apply concept activation factorizations on a layer $l' < l$ for an image containing a previously computed concept.

## 3.2   Ingredient 1: A Recursive Flavor

One of the most apparent issues in previous works [23, 24] is the choice of the layer at which the activation maps are extracted. Depending on this, certain concepts start getting amalgamated [44] into one, resulting in incoherent and indecipherable clusters, as illustrated in Fig 2. We posit that this can be solved by iteratively applying our decomposition at different layer-depths, and for the concepts that remain difficult to understand, look for their sub-concepts at earlier layers by isolating the images that contain them. This allows us to build hierarchies of concepts for each class.

We offer a simple solution consisting of reapplying our method to a concept by performing a second step of Concept Activation Factorization on a set of points that contain the concept $\mathcal{C}$ in order to refine it and create sub-concepts (e.g., decompose $\mathcal{C}$ into $\{\mathcal{C}_1, \mathcal{C}_2, \mathcal{C}_3\}$) see see Fig.2 for an illustrative example. Note that we generalize current methods in the sense that taking points $(x_1, ..., x_n)$ that are clustered in the logits layer (belonging to the same class) and decomposing them in a previous layer – as done in [23, 24] – is a valid recursive step. For a more general case, let us assume that a set of points that contain a common concept is obtained using a first step of Concept Activation Factorization. We then look for a set of points with a high coefficient for the concept of our choice to perform the next factorization. Formally, with a factorization for a layer $l$ $UW^T$ and a concept index $i$, this set of points is defined as $\mathcal{C} = \{\tau(x_j) : U(A_j)_i \geq \lambda\}$ In practice, we assume $\lambda$ to be equal to the 90th percentile of the values of $U_i$. Given this new set of points, we can then re-apply the Concept Matrix Factorization method to a earlier layer $l'$ – with $l' < l$ – to obtain the sub-concept's decomposition from the initial concept.

### 3.3 Ingredient 2: Sobol indices for enhanced concept importance estimation

A common concern with concept extraction methods is that what makes sense to humans is not necessarily what is being used by the model to predict. To avoid this kind of confirmation bias during our concept analysis phase, we can estimate the global importance of the extracted concepts. To do so, [22] proposed an estimator based on directional derivatives: the partial derivative of the model output with respect to the concept vector. While this measure is theoretically well founded, it relies on the same principle as gradient-based methods, and thus, suffers from the same pitfalls: neural network models have noisy gradients [5, 7]. Hence, the farther the chosen layer is from the output, the noisier the directional derivative score will be.

Since we essentially want to know which concept has the greatest effect on the output of the model, it is natural to consider the field of Sensitivity Analysis [45, 46, 32, 47]. In this section, we briefly recall the classical total Sobol indices and how to apply it to our problem. The complete derivation of the Sobol-Hoeffding decomposition is presented in the appendix D.

Formally, we place ourselves at layer $l$ and perform our Concept Activation Factorization, providing us with $U, W$. A natural way to estimate the importance of a concept $U_i$ is to measure the fluctuation of the model's output $h_{l,k}(UW^T)$ in response to meaningful perturbations of the concept coefficient $U_i$. Concretely, with $M = (M_1, ..., M_r) \in [0, 1]^r$, here an i.i.d sequence of real-valued random variables, we introduce a concept fluctuation to reconstruct a perturbated activation $\tilde{A} = (U \odot M)W^T$ (e.g., the masks can be used to put a concept value to zero). We can then propagate this perturbated activation to the model output $Y = h_{l,k}(\tilde{A})$. Thus, an important concept will have a large variance on the model output while an unused concept will barely change it.

Finally, we can capture the importance that a concept might have as a main effect – along with its interactions with other concepts – on the model's output by calculating the expected variance that would remain if all the indices of the masks except the $M_i$ were to be fixed. This yields the general definition of the Total Sobol indices.

**Definition 3.1** (**Total Sobol indices**). *The total Sobol index $\mathcal{S}_{T_i}$, which measures the contribution of a concept $U_i$ as well as its interactions of any order with any other concepts to the model output variance, is given by:*

$$\mathcal{S}_{T_i} = \frac{\mathbb{E}_{M_{\sim i}}(\mathbb{V}_{M_i}(Y|M_{\sim i}))}{\mathbb{V}(Y)} = \frac{\mathbb{E}_{M_{\sim i}}(\mathbb{V}_{M_i}(h_{l,k}((U \odot M)W^T)|M_{\sim i}))}{\mathbb{V}((U \odot M)W^T)} \quad (2)$$

In a practical way, this index can be calculated efficiently [48, 49, 50, 51, 52], more details on the sampling (Quasi-Monte Carlo) and the estimator used are left in appendix D.

### 3.4 Ingredient 3: Unlocking *Concept Attribution Map*

Attribution methods are useful for determining the regions deemed important by the model for the decision, but they lack the information about what exactly triggered it. We have seen that we can already extract this information from the matrices $U$ and $W$, but as it is, we cannot know to which part of the image the model associates each concept, and thus, better comprehend the model's decisions. In this section, we will show how we can unlock the set of attribution methods (forward and backward mode) to find where a concept is located in the input image (see Fig.1). Forward attribution methods don't rely on gradients and only use inference information, whereas backward methods require to back-propagate through the network's layers. By application of the chain rule, computing $\frac{\partial U}{\partial x}$ requires access to $\frac{\partial U}{\partial A}$.

To do so, it could be tempting to solve the linear system $UW^T = A$. However, this problem is ill-posed since $W^T$ is low rank. A standard approach is to calculate the Moore-Penrose pseudo-inverse $(W^T)^+$, which solves rank deficient systems by looking at the minimum norm solution [53]. In practice $(W^T)^+$ is computed with the Singular Value Decomposition (SVD) of $W^T$. Unfortunately, SVD is also the solution to the *unstructured minimization* of $\frac{1}{2}\|A - UW^T\|_F^2$ by the Eckart–Young–Mirsky theorem [54]. Hence, the non negativity constraints – i.e $U \geq 0, W \geq 0$ – of the NMF are ignored, which prevents approaches based on solving $U^TW = A^T$ from succeeding. Other issues stem from the fact that the $U, W$ decomposition is generally not unique.

Our third contribution consists on tackling this problem to allow the use of attribution methods – i.e. *Concept Attribution Maps* – by proposing a strategy to differentiate through the NMF layer.

**Implicit differentiation of NMF layers** The NMF problem 1 is NP-hard [55], and it is not convex with respect to the input pair $(U, W)$. However, fixing the value of one of the two factors and

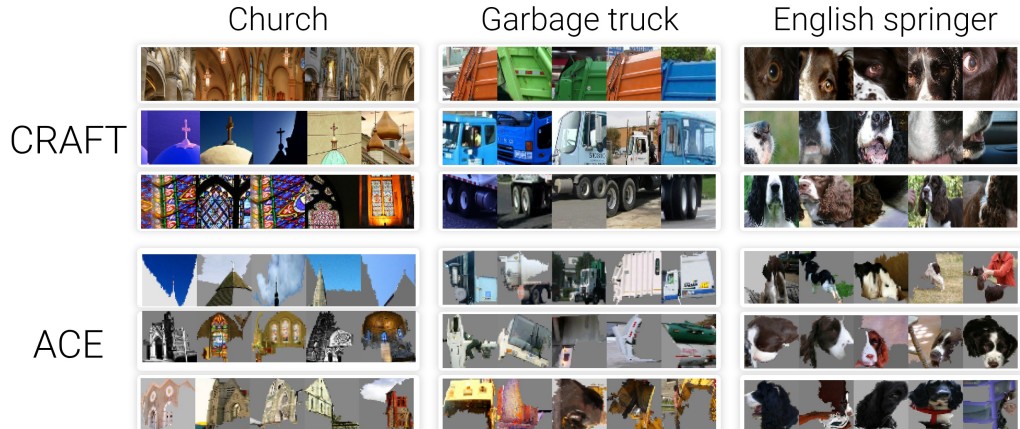

Figure 4: **Qualitative comparison.** We compare concepts found by our method (top) to those extracted with ACE [23] (bottom) for the classes *Church*, *Garbage truck* and *English springer* from ILSVRC2012 [27].

optimizing the other turns the NMF formulation into a pair of Non Negative Least Squares (NNLS) problems (see Equation 3), which are convex. This ensures that alternating minimization (a standard approach for NMF) of $(\boldsymbol{U}, \boldsymbol{W})$ factors will eventually reach a local (and global) minimum:

$$\boldsymbol{U}_{t+1} = \underset{\boldsymbol{U} \geq 0}{\arg\min} \frac{1}{2}\|\boldsymbol{A} - \boldsymbol{U}\boldsymbol{W}_t^T\|_F^2 \qquad \boldsymbol{W}_{t+1} = \underset{\boldsymbol{W} \geq 0}{\arg\min} \frac{1}{2}\|\boldsymbol{A} - \boldsymbol{U}_t\boldsymbol{W}^T\|_F^2 \qquad (3)$$

Each of the NNLS problem fulfills the KKT conditions[56, 57], which can be encoded in the so-called *optimality function* $\boldsymbol{F}$, see Equation 10 Appendix C.2. The implicit function theorem [39] allows us to use implicit differentiation [38, 39, 58] to efficiently compute the Jacobians $\frac{\partial \boldsymbol{U}}{\partial \boldsymbol{A}}$ and $\frac{\partial \boldsymbol{W}}{\partial \boldsymbol{A}}$ without requiring to back-propagate through each of the iterations of the NMF solver:

$$\frac{\partial(\boldsymbol{U}, \boldsymbol{W}, \bar{\boldsymbol{U}}, \bar{\boldsymbol{W}})}{\partial \boldsymbol{A}} = -(\partial_1 \boldsymbol{F})^{-1}\partial_2 \boldsymbol{F} \qquad (4)$$

However, this requires the dual variables $\bar{\boldsymbol{U}}$ and $\bar{\boldsymbol{W}}$, which are not computed by Scikit-learn's [59] popular implementation[†]. Consequently, we leverage the work of [62] and we re-implement our own solver with Jaxopt [40] based on ADMM [63], a GPU friendly algorithm (see Appendix C.2).

We start by performing Concept Activations Factorization – i.e we precompute the concept bank $\boldsymbol{W}$ by solving the NMF. Concept Attribution Maps of a new input $\boldsymbol{x}$ are calculated by solving the NNLS problem $\min_{\boldsymbol{U} \geq 0} \frac{1}{2}\|\boldsymbol{h}_l(\boldsymbol{x}) - \boldsymbol{U}\boldsymbol{W}^T\|_F^2$. The implicit differentiation of NMF layer $\frac{\partial \boldsymbol{U}}{\partial \boldsymbol{A}}$ is integrated into classical back-propagation to obtain $\frac{\partial \boldsymbol{U}}{\partial \boldsymbol{x}}$. Most interestingly, this technical advance unlocks all white-box explainability methods [5, 6, 7, 8, 9, 12] to generate concept-wise attribution maps and trace the part of the image that triggered the detection of the concept. Additionally, it is even possible to employ black-box methods [4, 13, 26, 14] since it only amounts to solving an NNLS problem.

## 4  Experimental evaluation

We used CRAFT to explain a ResNet50V2 trained on the ILSRVC2012 [27] data set (ImageNet). We selected a subset of 10 classes, each containing 1000 images (those recommended by ImageNette[‡]). In all of our experiments, $r = 25$, like in [23] and the cropping function $\boldsymbol{\tau}$ consists on randomly choosing 10 square $64 \times 64$ patches for each image. We start by qualitatively validating CRAFT by showing that: (1) the method yields concepts that are easy to interpret (see Fig. 4), (2) the combination of local and global explanations allows to explain complex failure cases otherwise unexplainable with only the attribution methods (see Fig. 5). Then, we validate independently the new ingredients brought by the method by showing quantitatively that (3) recursivity allows us to refine concepts, making them more meaningful to humans with the help of two psychophysics experiments, and (4) Sobol indices allow for a better estimation of concept importance. Additional experiments, including

---

[†]Scikit-learn uses a Block coordinate descent algorithm [60, 61], with a randomized SVD initialization.
[‡]https://github.com/fastai/imagenette

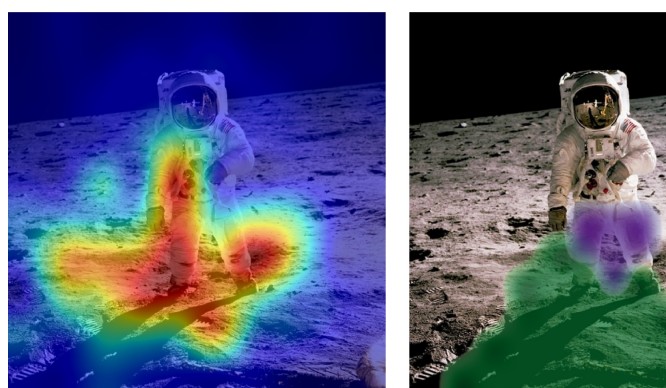

Figure 5: **This is a Shovel.** We compare a heatmap generated by RISE [13] (left) with the *Concept Attribution Maps* generated with our implicit differentiation pipeline and Grad-CAM (right) on the explanations of the two most influential concepts that drove the ResNet50's decision. We found a first concept that seems to be associated with textures of dirt commonly found in the images of the class *Shovel*. The second concept elucidated by CRAFT is located on the astronaut's pants, which he confuses with the ski suits of people clearing snow from their driveway with a shovel.

a sanity check and an example of activation maximization (Deep dream) on the concept bank, as well as many other examples of local explanations for randomly picked images from ILSVRC2012, are included in appendix B.

We leave a discussion on the limitations of this method and on the broader impact in appendix A.

## 4.1 Example of CRAFT concepts

Figure 4 compares the examples of concepts found by CRAFT against those found by ACE [23] for 3 classes of Imagenet. For each class the concepts are ordered by importance (the highest being the most important). ACE uses a clustering technique and TCAV to estimate importance, while CRAFT uses the method introduced in 3 and Sobol to estimate importance. These examples illustrate one of the weaknesses of ACE: the segmentation used can introduce biases through the baseline value used [64, 10]. The concepts found by CRAFT seem distinct: (vault, cross, stained glass) for the Church class, (dumpster, truck door, two-wheeler) for the garbage truck, and (eyes, nose, fluffy ears) for the English Springer. More examples can be found in the appendix.

## 4.2 Explaining complex failure cases

One of the goals of explainability is to explain the failure cases of the models studied. Figure 5 shows an example of an incorrect prediction: the model in question – here still a ResNet50 – predicts 'shovel'. Moreover, the attribution method on the left – here RISE [13] – does not tell us much except that the evidence for shovel seems to be located at the level of the ground and the lower torso and legs of the astronaut. With CRAFT, we can however study the concepts found by the model at these locations. There are two of them: the first concept in green, aims at the lunar ground and refers to the rocks often seen next to shovels in the dataset. The second concept in purple is aimed at the legs of the astronaut and refers to the legs of a person, often in a ski suit, which he takes for the astronaut's.

|  | Experts ($n = 36$) | Laymen ($n = 37$) |
|---|---|---|
| *Intruder* | | |
| Acc. Concept | 70.19% | 61.08% |
| Acc. Sub-Concept | 74.81% ($p = 0.18$) | **67.03**% ($p = 0.043$) |
| *Binary choice* | | |
| Sub-Concept | **76.1**% ($p < 0.001$) | **74.95**% ($p < 0.001$) |
| Odds Ratios | 3.53 | 2.99 |

Table 1: **Results from the psychopshysics experiments.**

## 4.3 Validation of Recursivity

To evaluate the meaningfulness of the extracted high-level concepts, we performed psychopshysic experiments with human subjects, to whom we requested to answer a survey in two phases. Furthermore, we distinguished two different audiences: on the one hand, experts in machine learning, and on

the other hand, people with no particular knowledge in computer vision. Both groups of participants were volunteers and didn't receive any monetary compensation. Some examples of the developed interface are available the appendix E.

**Intruder detection experiment,** we make users identify the intruder out of a series of five segments belonging to a certain class, with the odd one being taken from a different concept but from the same class. Now, we compare the results of this intruder detection with a concept (e.g., $\mathcal{C}_1$) coming from a layer $l$ and one of its sub-concepts (e.g., $\mathcal{C}_{12}$ in Fig.2) extracted using our recursive method. If the concept (or sub-concept) is meaningful, then it should be easy for the users to find the intruder. Table 1 summarizes our results, showing that indeed both concepts and sub-concepts are meaningful, and that recursivity can lead to a slightly higher understanding of the generated concepts (significant for non-experts, not significant for experts) and might suggest a way to make concepts more interpretable.

**Binary choice experiment,** In order to test the improvement of the meaningfulness of the sub-concept generated with recursivity with respect to the larger parent concept, we showed participants a segment belonging to a subcluster and to the parent cluster (e.g., $\boldsymbol{\tau}(\boldsymbol{x}) \subset \mathcal{C}_{11} \subset \mathcal{C}_1$) without specifying why those images are grouped together. We then we asked which of the two clusters (i.e., $\mathcal{C}_{11}$ or $\mathcal{C}_1$) seemed to accommodate the image the best. If our hypothesis is correct, then the concept refinement brought by recursivity should help form more coherent clusters. The results in Table 1 are satisfying, since in both the expert and non-expert groups, the participants chose the sub-cluster by more than 74% of the times. We measure the significance of our results by fitting a binomial logistic regression to our data, and we find that both groups are more likely to choose the sub-concept cluster (at a $p < 0.001$).

## 4.4 Fidelity analysis

We propose to simultaneously verify that the concepts are faithful to the model and that the concept importance estimator performs better than TCAV [22] by using the fidelity metrics introduced in [23, 24]. These metrics are similar to the one used for attribution methods, which consist on studying the change of the logit score when removing/adding pixels considered important. Nevertheless, we do not make these modifications in the pixel space but in the concept space: once $\boldsymbol{U}, \boldsymbol{W}$ are computed, we reconstruct the matrix $\boldsymbol{A} \approx \boldsymbol{U}\boldsymbol{W}^T$ us-ing only the most important concept (or

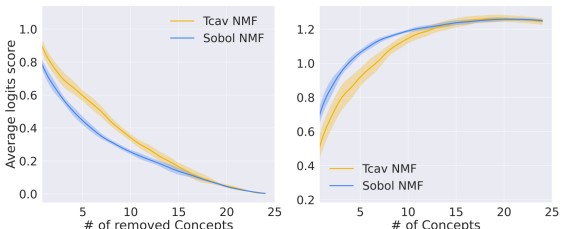

Figure 6: **(Left)** Deletion curve (lower is better). **(Right)** Insertion curves (higher is better). Whether in deletion or insertion, the score – calculated on more than 100,000 images – shows that using Sobol indices yield to better estimates of important concepts.

removing the most important concept for deletion), and study the score in output of the model. As can be seen from Fig. 6, ranking the extracted concepts using Sobol's importance score results in much steeper curves than when they are sorted by their TCAV scores. We confirm these results with other matrix factorization techniques (PCA, ICA, RCA) in the Appendix F.

## 5 Conclusion

In this paper, we introduced a method for automatically extracting human-scrutable concepts from Deep Neural Network: CRAFT. Our method allows to explain a pre-trained model both in a per-class and per-instance basis by highlighting both *what* the model saw when predicting the class label and *where* it is located, which, as we have shown, exhibits complementary benefits. The approach relies on three novel ingredients: 1) exploiting the recursive nature of the feature extraction chains in CNNs to find decompositions where each concept is clearly understandable; 2) measuring concept importance through Sobol indices to more accurately identify which concepts influence a model's decision for a given class; and 3) harnessing implicit differentiation to backpropagate through NMF blocks, thus enabling the use of any attribution method to create concept-wise local explanations that we call *Concept Attribution Maps*. Human experiments confirmed the validity of the approach and that concepts identified by CRAFT are meaningful. We hope that this work will guide further efforts in the search for concept-based explainability methods and that further connections between local and global explanations will be made.

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
