# A  Limitations and broader impact

## A.1  Limitations

Although we believe concept-based XAI to be a promising research direction, it isn't without pitfalls. It is capable of producing explanations that are ideally easy to understand by humans, but to what extent is a question that remains unanswered. The fact that there is no way to mathematically measure this prevents researchers from easily comparing the different techniques in the literature other than through time consuming and expensive experiments with human subjects. We think that developing a metric should be one of the field's priorities.

With CRAFT, we address the question of *what* by showing a cluster of the images that better represent each concept. However, we recognize that it's not perfect: in some cases, concepts are difficult to clearly define – put a label on what it represents –, and might induce some confirmation and selection bias. Feature visualization [18] might help in better illustrating the specific concept (as done in appendix B.3), but we believe there's still space for improvement. For instance, an interesting idea could be to leverage image captioning methods to describe the clusters of image crops, as textual information could help humans in better understanding clusters.

Although we believe CRAFT to be a considerable step in the good direction for the field of concept-based XAI, it also have some pitfalls. Namely, we chose the NMF as the activation factorization, which, while drastically improving the quality of extracted concepts, also comes with it's own caveats. For instance, it is known to be NP-hard to compute exactly, and in order to make it scalable, we had to use a tractable approximation by alternating the optimization of $U$ and $W$ through ADMM [63]. This approach might indeed yield non-unique solutions. Our experiments (section 4.4), have shown a low variance on between the runs, which comforts us about the stability of our results.However the absence of formal guarantee for uniqueness must be kept in mind: this subject is still an active topic of research and improvement could be expected in the near future. Namely, sparsity constraints and regularization seem to be promising paths. Naturally, we also need enough samples of the class under study to be available for the factorization to construct a relevant concept bank, which might affect the quality of the explanations on frugal applications where data is very scarce.

## A.2  Broader impact

We do hope that CRAFT helps in the transition to more human-understandable ways of explaining neural network models. It's capacity to find easily understandable concepts inside complex architectures and providing an indication of *where* they are located in the image is – to the best of our knowledge – unmatched. We also think that this method's structure is a step towards reducing confirmation bias: for instance dataset's labels are never used in this method, only the model's predictions. Without claiming to remove confirmation bias, the method focuses on what *the model sees* rather than what *we expect the model to see*. We believe this can help end-users build trust on computer vision models, and at the same time, provide ML practitioners with insights into potential sources of bias in the dataset (e.g. the ski pants in the astronaut/shovel example). Other methods in the literature obtaining similar results require very specific architectures [36] or to train another model to generate the explanations [66], so CRAFT provides a considerable advantage in the matter of flexibility in comparison.

# B  More results of CRAFT

## B.1  Concept Attribution Maps.

We show more examples of Concept Attribution Maps for the classes 'Chain saw' in Figure S2 and 'Parachute' in Figure S1.

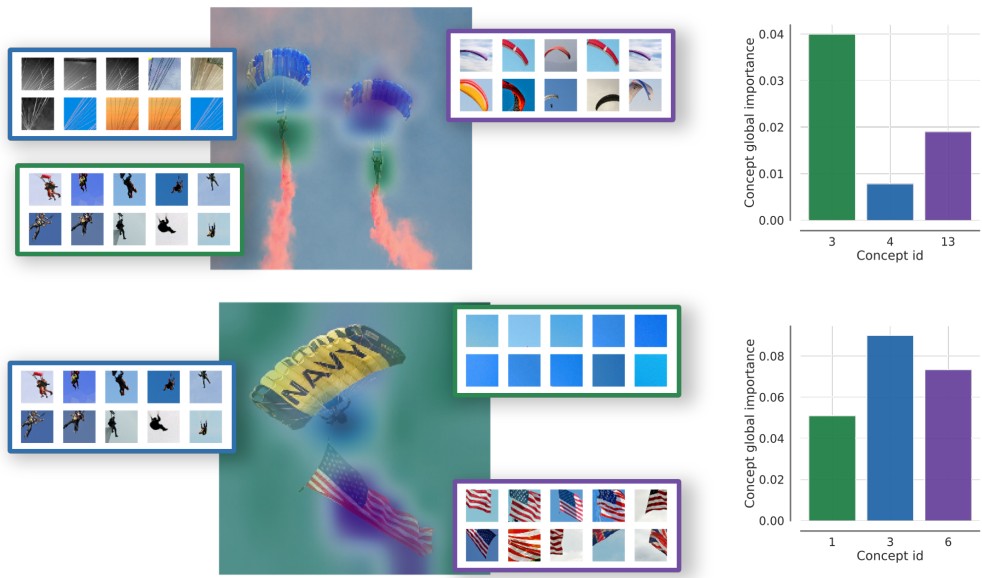

Figure S1: **CRAFT results for the class 'Parachute'**. The model under study is a ResNet50, and we used the penultimate layer to apply the matrix activation factorization.

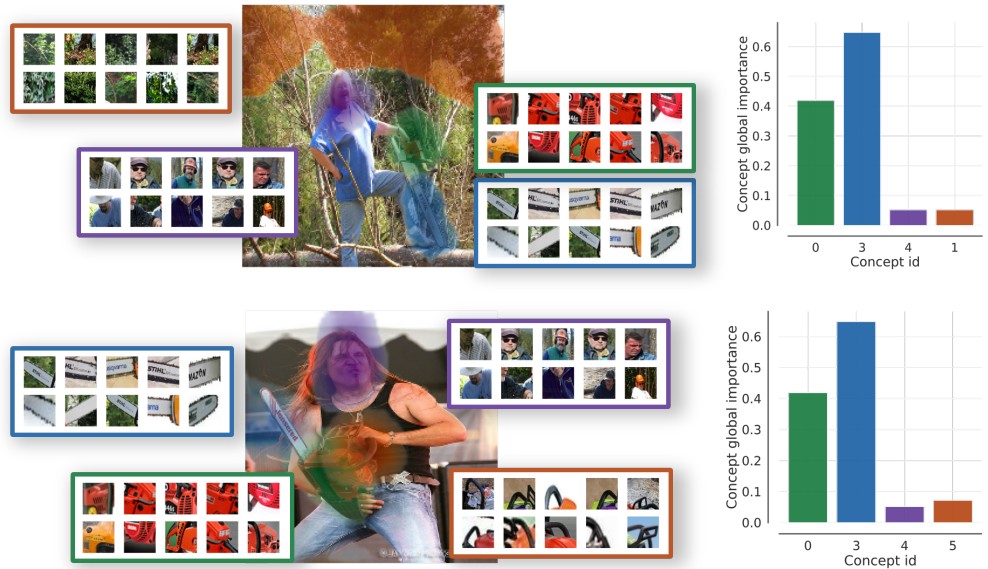

Figure S2: **CRAFT results for the class 'Chain saw'**. The model under study is a ResNet50 we used the penultimate layer to apply the matrix activation factorization.

## B.2 Most important concepts.

We show more example of the 4 most importants concepts for 6 classes: 'Chain saw' and 'English springer' (Figure S3), 'Gas pump' and 'Golf ball' (Figure S4), 'French horn' and 'Garbage Truck' (Figure S5).

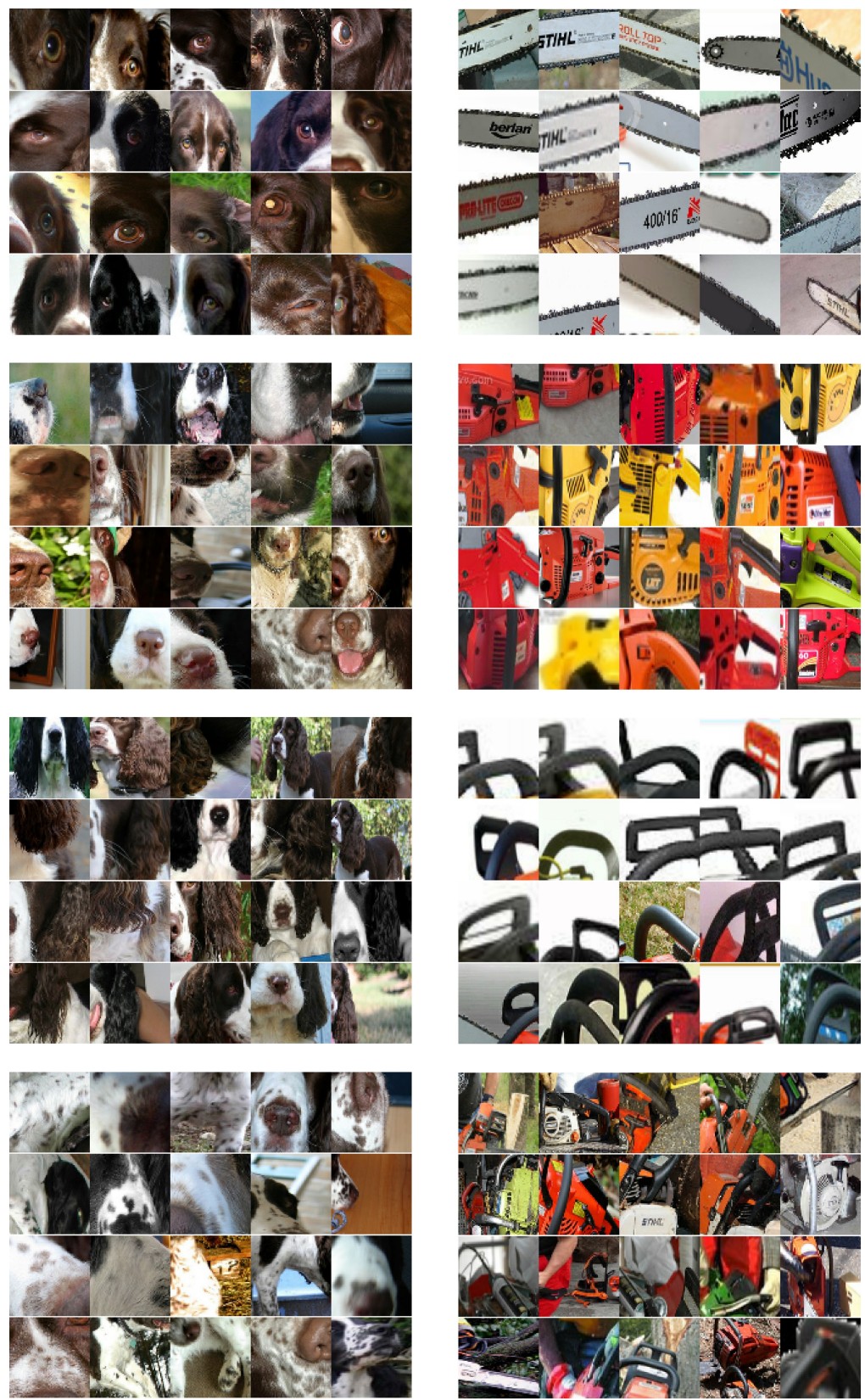

Figure S3: **CRAFT most important concepts**. The 4 most important concepts (higher means more important) for 'English springer' (left) and 'Chain saw' (right).

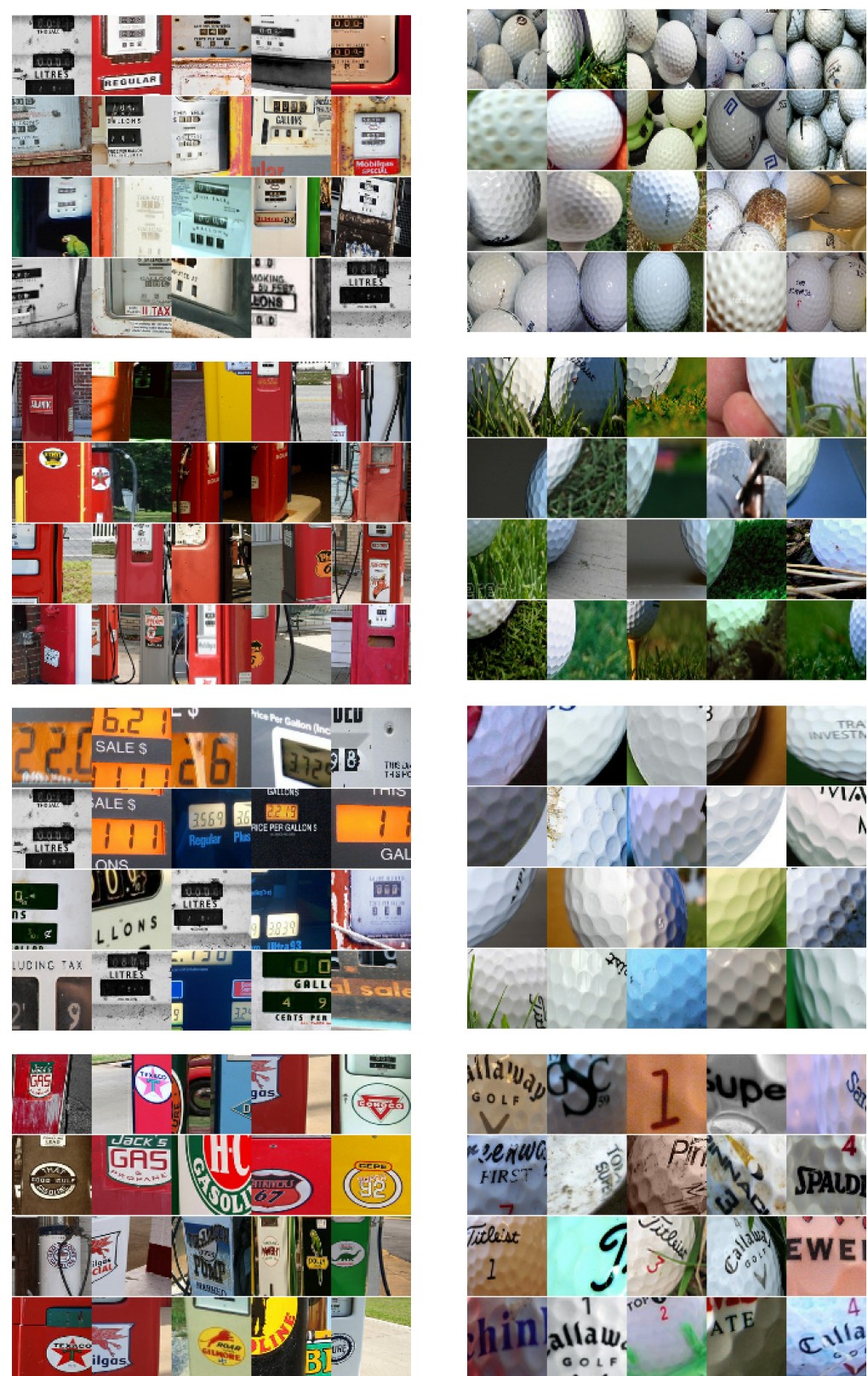

Figure S4: **CRAFT most important concepts**. The 4 most important concepts (higher means more important) for 'Gas pump' (left) and 'Golf ball' (right).

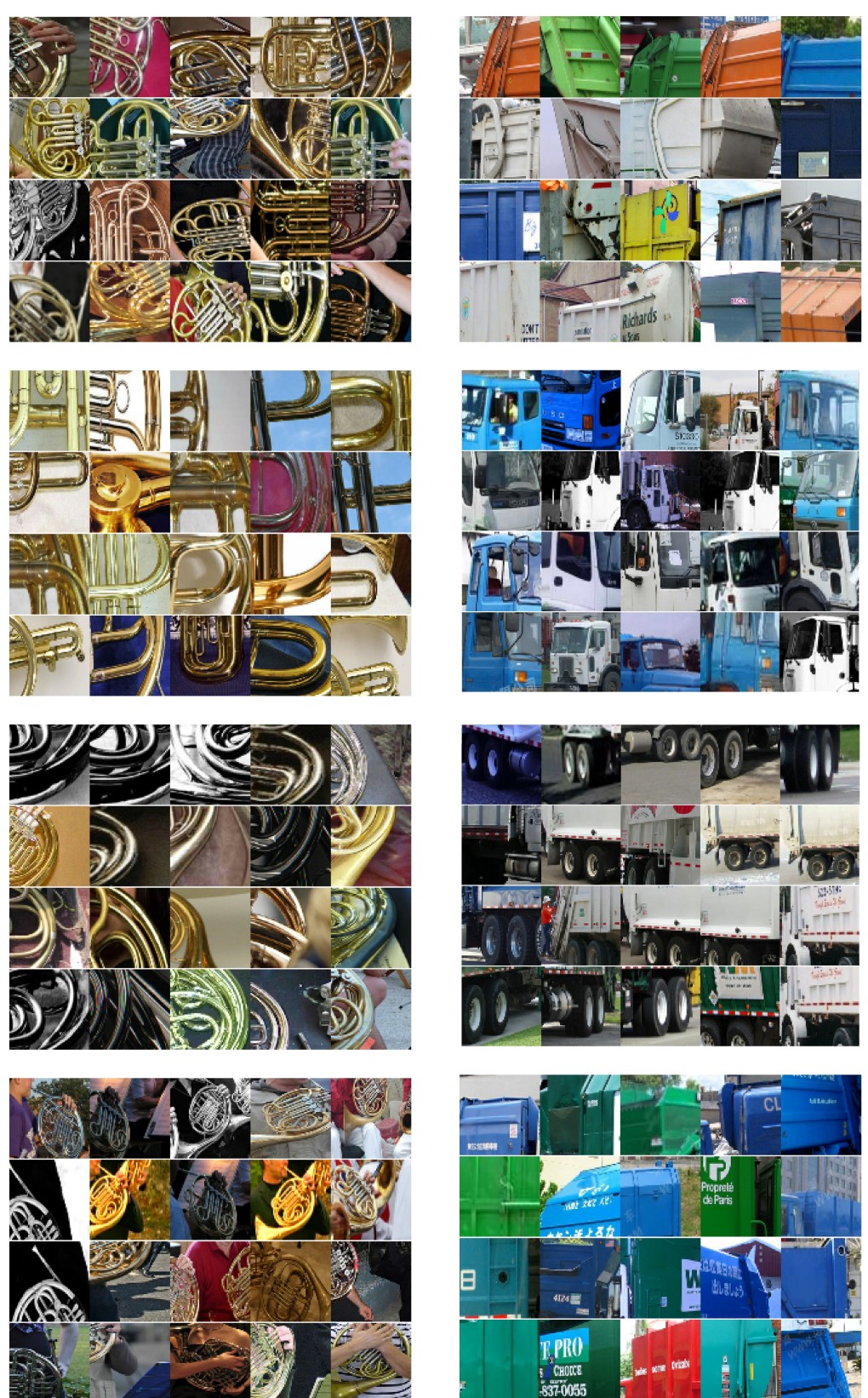

Figure S5: **CRAFT most important concepts**. The 4 most important concepts (higher means more important) for 'French horn' (left) and 'Garbage truck' (right).

## B.3 Feature Visualization validation

Another way of interpreting concepts – as per [22] – is to employ feature visualization methods: through optimization, find an image that maximizes an activation pattern. In our case, we used the set of regularization and constraints proposed by [18], which allow us to successfully obtain realistic images. In Figure S6, we showcase these synthetic images obtained through feature visualization, along with the segments that maximize the target concept. We observe that they do reflect the underlying concepts of interest.

Concretely, to produce those feature visualization, we are looking for an image $\boldsymbol{x}^*$ that is optimized to correspond to a concept from the concept bank $\boldsymbol{W}_i$. We use the so called 'dot-cossim' loss proposed by [18], which give the following objective:

$$\boldsymbol{x}^* = \underset{\boldsymbol{x} \in \mathcal{X}}{\arg\max} \ \langle \boldsymbol{h}_l(\boldsymbol{x}), \boldsymbol{W}_i \rangle \frac{\langle \boldsymbol{h}_l(\boldsymbol{x}), \boldsymbol{W}_i \rangle^2}{||\boldsymbol{h}_l(\boldsymbol{x})|| \ ||\boldsymbol{W}_i||} - \mathcal{R}(\boldsymbol{x})$$

With $\mathcal{R}(\cdot)$, the regularizations applied to $\boldsymbol{x}$ – the default regularizations in the **Xplique** library [67]. As for the specific parameters, we used Fourier preconditioning on the image with a decay rate of $0.8$ and an Adam optimizer ($lr = 1e - 1$).

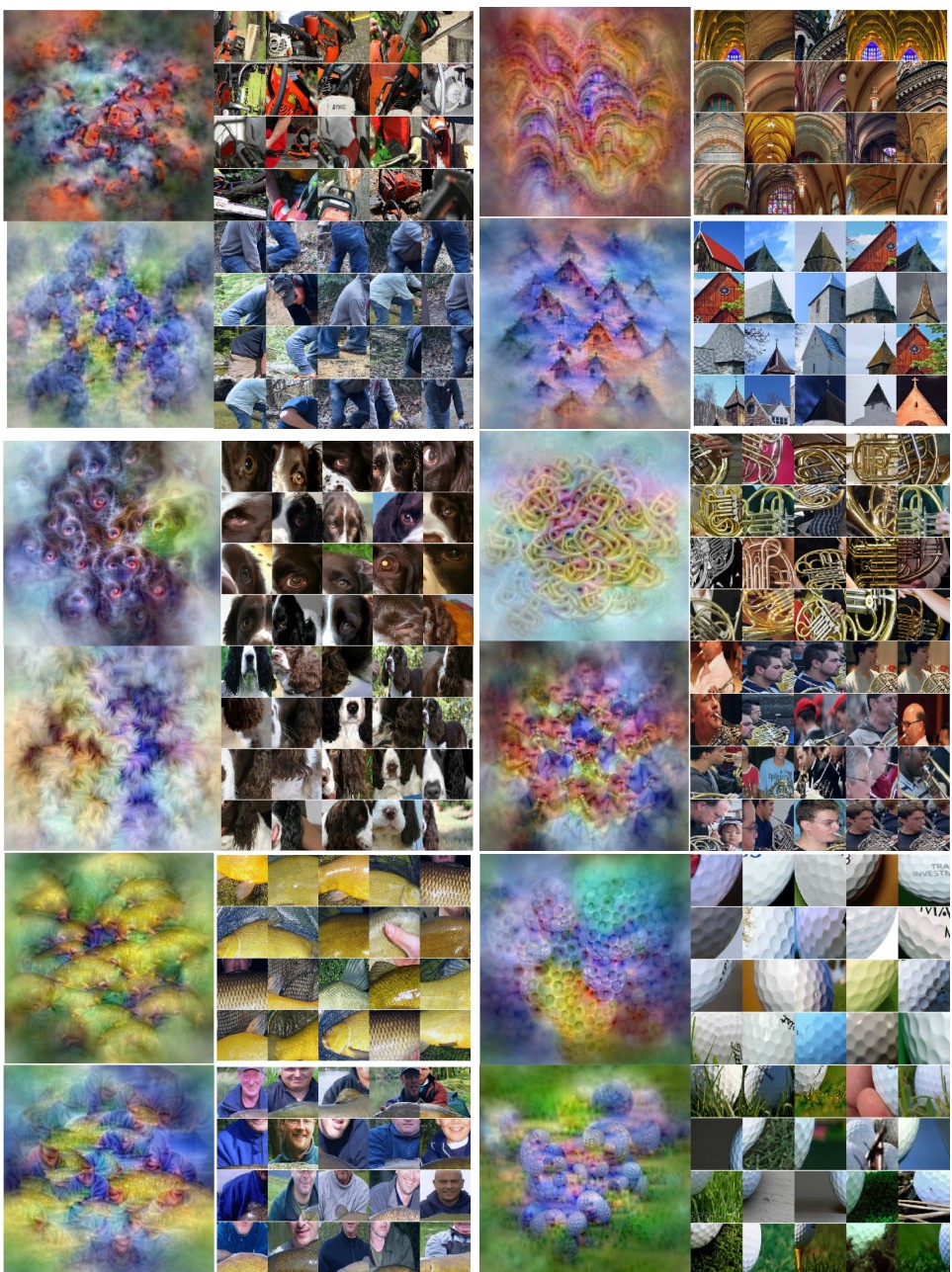

Figure S6: **Feature visualization for CRAFT concepts.**. The model under study is a ResNet50 we used the penultimate layer to apply the matrix activation factorization.

 ## C   Backpropagating through the NMF block

 ### C.1   Alternating Direction Method of Multipliers (ADMM) for NMF

We recall that NMF decomposes the positive features vector $\boldsymbol{A} \in \mathbb{R}^{n \times p}$ of $n$ examples lying in dimension $p$, into a product of positive low rank matrices $\boldsymbol{U}(\boldsymbol{A}) \in \mathbb{R}^{n \times r}$ and $\boldsymbol{W}(\boldsymbol{A}) \in \mathbb{R}^{p \times r}$ (with $r << \min(n, p)$), i.e the solution to the problem:

$$\min_{\boldsymbol{U} \geq 0, \boldsymbol{W} \geq 0} \frac{1}{2} \|\boldsymbol{A} - \boldsymbol{U}\boldsymbol{W}^T\|_F^2 \tag{5}$$

For simplicity we used a non-regularized version of the NMF objective, following Algorithms 1 and 3 in paper [62], based on ADMM [63]. This algorithm transforms the non-linear equality constraints into indicator functions $\delta$. Auxiliary variables $\tilde{U}, \tilde{W}$ are also introduced to separate the optimization of the objective on the one side, and the satisfaction of the constraint on $U, W$ on the other side. The equality constraints $\tilde{U} = U, \tilde{W} = W$ are linear and easily handled by the ADMM framework through the associated dual variables $\bar{U}, \bar{W}$. In our case, the problem in Equation 5 is transformed into:

$$\min_{U, \tilde{U}, W, \tilde{W}} \frac{1}{2} \|A - \tilde{U}\tilde{W}^T\|_F^2 + \delta(U) + \delta(W)$$
$$s.t. \ \tilde{U} = U, \tilde{W} = W \tag{6}$$
$$\text{with } \delta(H) = \begin{cases} 0 \text{ if } H \geq 0 \\ +\infty \text{ otherwise} \end{cases}$$

The constraints are simplified at the cost of a non-smooth (and even a non-finite) objective function $\frac{1}{2}\|A - \bar{U}\bar{W}^T\|_F^2 + \delta(U) + \delta(W)$ due to the term $\delta(U) + \delta(W)$. ADMM proceeds to create a so-called *augmented Lagrangian* with $l_2$ regularization $\rho > 0$:

$$\mathcal{L}(A, U, W, \tilde{U}, \tilde{W}, \bar{U}, \bar{W}) = \frac{1}{2}\|A - \tilde{U}\tilde{W}^T\|_F^2 + \delta(U) + \delta(W)$$
$$+ \bar{U}^T(\tilde{U} - U) + \bar{W}^T(\tilde{W} - W) \tag{7}$$
$$+ \frac{\rho}{2}\left(\|\tilde{U} - U\|_2^2 + \|\tilde{W} - W\|_2^2\right)$$

The (regularized) problem associated to this Lagrangian is decomposed into a sequence of convex problems that alternate minimization over the $U, \tilde{U}, \bar{U}$ and the $W, \tilde{W}, \bar{W}$ triplets.

$$U_{t+1} = \arg\min_{U = \tilde{U}} \frac{1}{2}\|A - \tilde{U}W_t^T\|_F^2 + \delta(U) + \frac{\rho}{2}\|\tilde{U} - U\|_2^2 \tag{8}$$

$$W_{t+1} = \arg\min_{W = \tilde{W}} \frac{1}{2}\|A - U_t\tilde{W}^T\|_F^2 + \delta(W) + \frac{\rho}{2}\|\tilde{W} - W\|_2^2 \tag{9}$$

This guarantees a monotonic decrease of the objective function $\|A - \tilde{U}_t\tilde{W}_t^T\|_F^2$. Each of these sub-problems is thus solved with ADMM separately, by alternating minimization steps of $\frac{1}{2}\|A - \tilde{U}W_t^T\|_F^2 + \bar{U}^T(\tilde{U} - U) + \frac{\rho}{2}\|U - \tilde{U}\|_2^2$ over $\tilde{U}$ (*i*), with minimization steps of $\delta(U) + \frac{\rho}{2}\|U - \tilde{U}\|_2^2$ over $U$ (*ii*), and gradient ascent steps (*iii*) on the dual variable $\bar{U} \leftarrow \bar{U} + (\tilde{U} - U)$. A similar scheme is used for $W$ updates. Step (*i*) is a simple convex quadratic program with equality constraints, whose KKT [56, 57] conditions yield a linear system with a Positive Semi-Definite (PSD) matrix. Step (*ii*) is a simple projection of $\tilde{U}$ onto the convex set $\delta^{-1}(0)$. Finally, step (*iii*) is inexpensive.

Concretely, we solved the quadratic program using Conjugate Gradient [68], from *jax.scipy.sparse.linalg.cg*. This indirect method only involves *matrix-vector* products and can be more GPU-efficient than methods that are based on matrix factorization (such as Cholesky decomposition). Also, we re-implemented the pseudo code in [62] in *Jax* for a fully GPU-compatible program. We used the primal variables $U_0, W_0$ returned by *sklearn.decompose.nmf* as a *warm start* for ADMM and observe that the high quality initialization of these primal variables considerably speeds up the convergence of the dual variables.

## C.2 Implicit differentiation

The Lagrangian of the NMF problem reads $\mathcal{L}(U, W, \bar{U}, \bar{W}) = \frac{1}{2}\|A - UW^T\|_F^2 - \bar{U}^TU - \bar{W}^TW$, with dual variables $\bar{U}$ and $\bar{W}$ associated to the constraints $U \geq 0, W \geq 0$. It yields a function $F$ based on the KKT conditions [56, 57] whose optimal tuple $U, W, \bar{U}, \bar{W}$ is a root.

For single NNLS problem (for example, with optimization over $U$) the KKT conditions are:

$$\begin{cases} \nabla_U \left( \frac{1}{2}\|\boldsymbol{A} - \tilde{\boldsymbol{U}}\tilde{\boldsymbol{W}}^T\|_F^2 + \bar{\boldsymbol{U}}^T(-\boldsymbol{U}) \right) = 0, & \text{stationarity} \\ -\boldsymbol{U} \leq 0, & \text{primal feasability} \\ \bar{\boldsymbol{U}} \odot \boldsymbol{U} = 0, & \text{complementary slackness} \\ \bar{\boldsymbol{U}} \geq 0, & \text{dual feasability} \end{cases} \tag{10}$$

By stacking the KKT conditions of the NNLS problems the we obtain the so-called *optimality function* $\boldsymbol{F}$:

$$\boldsymbol{F}((\boldsymbol{U},\boldsymbol{W},\bar{\boldsymbol{U}},\bar{\boldsymbol{W}}),\boldsymbol{A}) = \begin{cases} (\boldsymbol{U}\boldsymbol{W}^T - \boldsymbol{A})\boldsymbol{W} - \bar{\boldsymbol{U}}, \\ (\boldsymbol{W}\boldsymbol{U}^T - \boldsymbol{A}^T)\boldsymbol{U} - \bar{\boldsymbol{W}}, \\ \bar{\boldsymbol{U}} \odot \boldsymbol{U}, \\ \bar{\boldsymbol{W}} \odot \boldsymbol{W}, \end{cases} \tag{11}$$

The implicit function theorem [39] allows us to use implicit differentiation [38, 39, 58] to efficiently compute the Jacobians $\frac{\partial \boldsymbol{U}}{\partial \boldsymbol{A}}$ and $\frac{\partial \boldsymbol{W}}{\partial \boldsymbol{A}}$ without requiring to back-propagate through each of the iterations of the NMF solver:

$$\frac{\partial(\boldsymbol{U},\boldsymbol{W},\bar{\boldsymbol{U}},\bar{\boldsymbol{W}})}{\partial \boldsymbol{A}} = -(\partial_1 \boldsymbol{F})^{-1}\partial_2 \boldsymbol{F} \tag{12}$$

Implicit differentiation requires access to the dual variables of the optimization problem in equation 1, which are not computed by Scikit-learn's popular implementation. Scikit-learn uses Block coordinate descent algorithm [60, 61], with a randomized SVD initialization. Consequently, we leverage our implementation in Jax based on ADMM [63].

Concretely, we perform a two-stage backpropagation *Jax (2)↝Tensorflow (1)* to leverage the advantage of each framework. The lower stage (1) corresponds to feature extraction $\boldsymbol{A} = \boldsymbol{h}_l(\boldsymbol{X})$ from crops of images $\boldsymbol{X}$, and upper stage (2) computes NMF $\boldsymbol{A} \approx \boldsymbol{U}\boldsymbol{W}^T$.

We use the *Jaxopt* [40] library that allows efficient computation of $\frac{\partial(\boldsymbol{U},\boldsymbol{W},\bar{\boldsymbol{U}},\bar{\boldsymbol{W}})}{\partial \boldsymbol{A}} = -(\partial_1 \boldsymbol{F})^{-1}\partial_2 \boldsymbol{F}$. The matrix $(\partial_1 \boldsymbol{F})^{-1}$ is never explicitly computed – that would be too costly. Instead, the system $\partial_1 \boldsymbol{F}\frac{\partial(\boldsymbol{U},\boldsymbol{W},\bar{\boldsymbol{U}},\bar{\boldsymbol{W}})}{\partial \boldsymbol{A}} = -\partial_2 \boldsymbol{F}$ is solved with Conjugate Gradient [68] through the use of Jacobian Vector Products (JVP) $\boldsymbol{v} \mapsto (\partial_1 \boldsymbol{F})\boldsymbol{v}$.

The chain rule yields:

$$\frac{\partial \boldsymbol{U}}{\partial \boldsymbol{X}} = \frac{\partial \boldsymbol{A}}{\partial \boldsymbol{X}} \frac{\partial \boldsymbol{U}}{\partial \boldsymbol{A}}$$

Usually, most Autodiff frameworks (e.g Tensorflow, Pytorch, Jax) handle it automatically. Unfortunately, combining two of those framework raises a new difficulty since they are not compatible. Hence, we re-implement manually the two stages auto-differentiation.

Since $r$ is far smaller ($r = 25$ in all our experiments) than input dimension $\boldsymbol{X}$ (typically $224 \times 244$ for ImageNet images), back-propagation is the preferred algorithm in this setting over forward-propagation. We start by computing sequentially the gradients $\nabla_{\boldsymbol{X}}\boldsymbol{U}_i$ for all concepts $1 \leq i \leq r$. This amounts to compute $\boldsymbol{v} = \nabla_{\boldsymbol{A}}\boldsymbol{U}_i$ with Implicit Differentiation in Jax, convert the Jax array $\boldsymbol{v}$ into Tensorflow tensor, and then to compute $\nabla_{\boldsymbol{X}}\boldsymbol{U}_i = \frac{\partial \boldsymbol{A}}{\partial \boldsymbol{X}}\nabla_{\boldsymbol{A}}\boldsymbol{U}_i = \nabla_{\boldsymbol{X}}(\boldsymbol{h}_l(\boldsymbol{X}) \cdot \boldsymbol{v})$. The latter is easily done in Tensorflow. Finally we stack the gradients $\nabla_{\boldsymbol{X}}\boldsymbol{U}_i$ to obtain the Jacobian $\frac{\partial \boldsymbol{U}}{\partial \boldsymbol{X}}$.

## D   Sobol indices for concepts

We propose to formally derive the Sobol indices for the estimation of the importance of concepts. Let us define a probability space $(\Omega, \mathcal{A}, \mathbb{P})$ of possible concept perturbations. In order to build these concept perturbations, we start from an original vector of concepts coefficient $\widehat{\boldsymbol{U}} \in \mathbb{R}^r$ and use stochastic masks $\boldsymbol{M} = (M_1, ..., M_r) \in \mathcal{M} \subseteq [0,1]^r$, as well as a perturbation operator $\boldsymbol{\pi}$ : $\mathcal{A} \times \mathcal{M} \to \mathcal{A}$ to create stochastic perturbation of $\widehat{\boldsymbol{U}}$ that we call concept perturbation $\boldsymbol{U} = \boldsymbol{\pi}(\widehat{\boldsymbol{U}}, \boldsymbol{M})$.

Concretely, to create our concept perturbation we consider the inpainting function as our perturbation operator (as in [4, 13, 14]) : $\boldsymbol{\pi}(\tilde{\boldsymbol{U}}, \boldsymbol{M}) = \tilde{\boldsymbol{U}} \odot \boldsymbol{M} + (1 - \boldsymbol{M})\mu$ with $\odot$ the Hadamard product

and $\mu \in \mathbb{R}$ a baseline value, here zero. For the sake of notation, we will note $\boldsymbol{f} : \mathcal{A} \to \mathbb{R}$ the function mapping a random concept perturbation $\boldsymbol{U}$ from the layer $l$ to the output. We denote the set $\mathcal{U} = \{1, ..., r\}$, $\boldsymbol{u}$ a subset of $\mathcal{U}$, its complementary $\sim \boldsymbol{u}$ and $\mathbb{E}(\cdot)$ the expectation over the perturbation space. Finally, we assume that $\boldsymbol{f} \in \mathbb{L}^2(\mathcal{A}, \mathbb{P})$ i.e. $|\mathbb{E}(\boldsymbol{f}(\boldsymbol{U}))| < +\infty$.

The Hoeffding decomposition allows us to express the function $\boldsymbol{f}$ into summands of increasing dimension, denoting $\boldsymbol{f_u}$ the partial contribution of the concepts $\boldsymbol{U_u} = (U_i)_{i \in \boldsymbol{u}}$ to the score $\boldsymbol{f}(\boldsymbol{U})$:

$$
\begin{aligned}
\boldsymbol{f}(\boldsymbol{U}) &= \boldsymbol{f}_\varnothing + \sum_i^d \boldsymbol{f}_i(U_i) + \sum_{1 \leqslant i < j \leqslant d} \boldsymbol{f}_{i,j}(U_i, U_j) + \cdots + \boldsymbol{f}_{1,...,r}(U_1, ..., U_r) \\
&= \sum_{\boldsymbol{u} \subseteq \mathcal{U}} \boldsymbol{f_u}(\boldsymbol{U_u})
\end{aligned}
\tag{13}
$$

Eq. 13 consists of $2^r$ terms and is unique under the following orthogonality constraint:

$$
\forall (\boldsymbol{u}, \boldsymbol{v}) \subseteq \mathcal{U}^2 \; s.t. \; \boldsymbol{u} \neq \boldsymbol{v}, \;\; \mathbb{E}\big(\boldsymbol{f_u}(\boldsymbol{U_u})\boldsymbol{f_v}(\boldsymbol{U_v})\big) = 0
\tag{14}
$$

Furthermore, orthogonality yields the characterization $\boldsymbol{f_u}(\boldsymbol{U_u}) = \mathbb{E}(\boldsymbol{f}(\boldsymbol{U})|\boldsymbol{U_u}) - \sum_{\boldsymbol{v} \subset \boldsymbol{u}} \boldsymbol{f_v}(\boldsymbol{U_v})$ and allows us to decompose the model variance as:

$$
\begin{aligned}
\mathbb{V}(\boldsymbol{f}(\boldsymbol{U})) &= \sum_i^d \mathbb{V}(\boldsymbol{f}_i(U_i)) + \sum_{1 \leqslant i < j \leqslant d} \mathbb{V}(\boldsymbol{f}_{i,j}(U_i, U_j)) + ... + \mathbb{V}(\boldsymbol{f}_{1,...,r}(U_1, ..., U_r)) \\
&= \sum_{\boldsymbol{u} \subseteq \mathcal{U}} \mathbb{V}(\boldsymbol{f_u}(\boldsymbol{U_u}))
\end{aligned}
\tag{15}
$$

Building from Eq. 15, it is natural to characterize the influence of any subset of concepts $\boldsymbol{u}$ as its own variance w.r.t. the total variance. This yields, after normalization by $\mathbb{V}(\boldsymbol{f}(\boldsymbol{U}))$, the general definition of Sobol' indices.

**Definition D.1** (Sobol indices [46]). The sensitivity index $\mathcal{S}_{\boldsymbol{u}}$ which measures the contribution of the concept set $\boldsymbol{U_u}$ to the model response $\boldsymbol{f}(\boldsymbol{U})$ in terms of fluctuation is given by:

$$
\mathcal{S}_{\boldsymbol{u}} = \frac{\mathbb{V}(\boldsymbol{f_u}(\boldsymbol{U_u}))}{\mathbb{V}(\boldsymbol{f}(\boldsymbol{U}))} = \frac{\mathbb{V}(\mathbb{E}(\boldsymbol{f}(\boldsymbol{U})|\boldsymbol{U_u})) - \sum_{\boldsymbol{v} \subset \boldsymbol{u}} \mathbb{V}(\mathbb{E}(\boldsymbol{f}(\boldsymbol{U})|\boldsymbol{U_v}))}{\mathbb{V}(\boldsymbol{f}(\boldsymbol{U}))}
\tag{16}
$$

Sobol indices give a quantification of the importance of any subset of concepts with respect to the model decision, in the form of a normalized measure of the model output deviation from $\boldsymbol{f}(\boldsymbol{U})$. Thus, Sobol indices sum to one : $\sum_{\boldsymbol{u} \subseteq \mathcal{U}} \mathcal{S}_{\boldsymbol{u}} = 1$.

Furthermore, the framework of Sobol' indices enables us to easily capture higher-order interactions between features. Thus, we can view the Total Sobol indices defined in 2 as the sum of of all the Sobol indices containing the concept $i$ : $\mathcal{S}_{T_i} = \sum_{\boldsymbol{u} \subseteq \mathcal{U}, i \in \boldsymbol{u}} \mathcal{S}_{\boldsymbol{u}}$. Concretely, we estimate the total Sobol indices using the Jansen estimator [50] and Quasi-Monte carlo Sequence (Sobol $LP_\tau$ sequence).

# E   Human experiments

We first describe how participants were enrolled in the study, then our general experimental design (See SI for more informations).

**Participants**   Behavioral accuracy data were gathered from $n = 73$ participants. All participants provided informed consent electronically in order to perform the experiment ($\sim 4 - 6$ min). The protocol was approved by the University IRB and was carried out in accordance with the provisions of the World Medical Association Declaration of Helsinki. For each of the 2 experiment tested, we had prepared filtering criteria for uncooperative people (namely based on time), but all participants passed these filters.

**General study design**    For the first experiment – consisting in finding the intruder among elements of the same concept and an element from a different concept (but of the same class, see Figure S8) – the choice was randomized in order to avoid any kind of bias due to the order of presentation of the choices. Moreover, in order to avoid any bias coming from the participants themselves (one group being more successful than the other) all participants were in the two conditions of finding intruders in batches of images coming from either concepts or sub-concepts. Concerning experiment 2, the order was also randomized (see see Figure S9).

The participants had to successively find 30 intruders (15 block concepts and 15 block sub-concepts) for experiment 1 and then make 15 choices (sub-concept vs concept) for experiment 2, see Figure S7.

The expert participants are people working in machine learning (researchers, software developers, engineers) and have participated in the study following an announcement in the authors' laboratory/company. The other participants (Laymen) have no particular competence in machine learning.

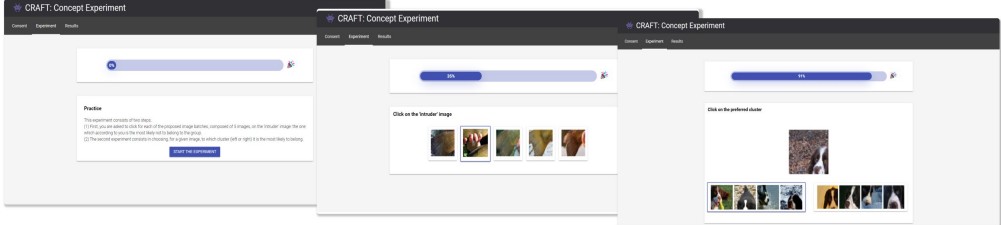

Figure S7: **Human Experiment Website.**

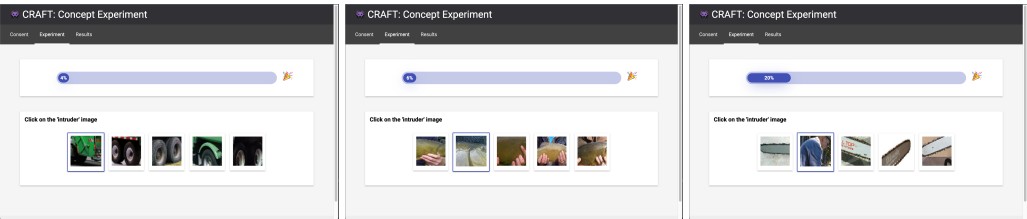

Figure S8: **Binary choice experiment.**

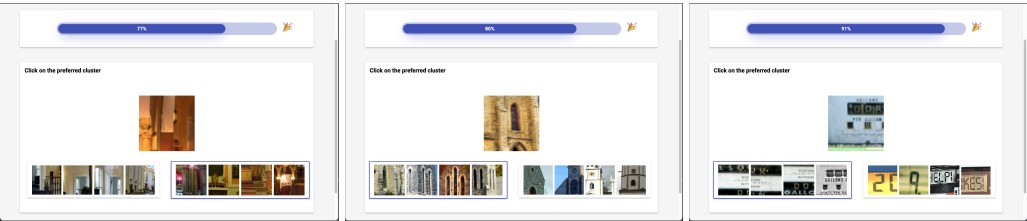

Figure S9: **Intruder experiment.**

# F    Fidelity experiments

For our experiments on the concept importance measure, we focused on certain classes of IL-SRVC2012 [27] and used a ResNet50V2 [69] that had already been trained on this dataset. Just like in [23, 24], we measure the insertion and deletion metrics for our concept extraction technique – as well as concepts vectors extracted using PCA, ICA and RCA as dimensionality reduction algorithms, see Figure S10 – and we compare them when we add/remove the concepts as ranked by the TCAV score [22] and by the Sobol importance score. As originally explained in [13], the objective of these metrics is to add/remove parts of the input according to how much an explainability method considers that it is influential and looking at the speed at which the logit for the predicted class increases/decreases.

In particular, for our experimental evaluations, we have randomly chosen 100000 images from ILSVRC2012 [27] and computed the deletion and insertion metrics for 5 different seeds – for a total of half a million images. In Figure S10, the shade around the curves represent the standard deviation over these 5 experiments.

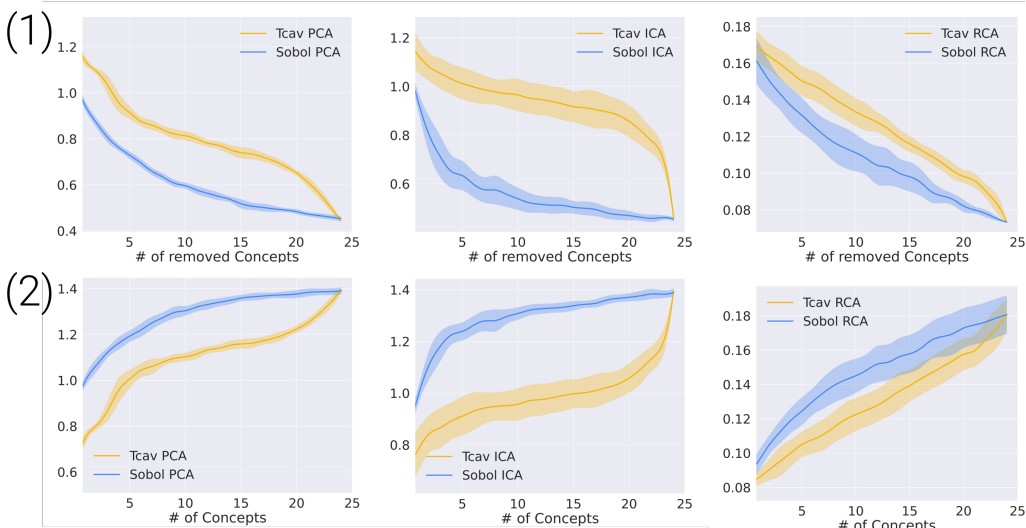

Figure S10: **(1) Deletion curves** for different concept extraction methods, Sobol outperforms TCAV not only for NMF to correctly estimate concept importance (lower is better). **(2) Insertion curves** for different concept extraction methods, Sobol outperforms TCAV to correctly estimate concept importance (higher is better).

## G Additional examples of concepts and sub-concepts

### G.1 Sanity Check

Following the work from [30], we performed a sanity check on our method, by running the concept extraction pipeline on a randomized model. This procedure was performed on a ResNet-50v2 model with randomized weights. When weights are randomized, concepts are mainly based on color histograms. This might result from skip connections which propagate signal from the inputs.

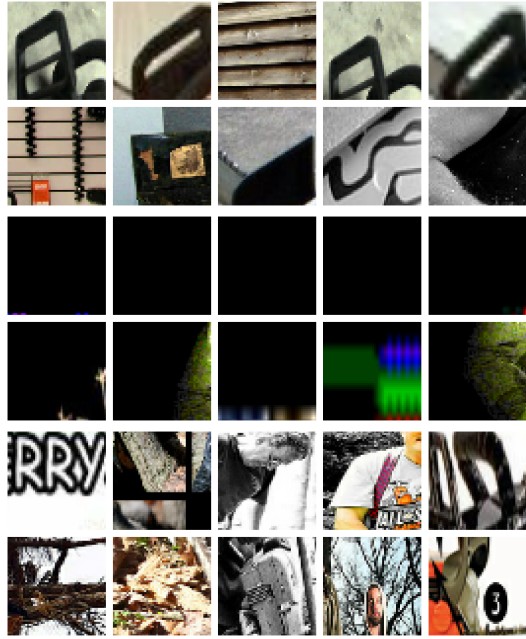

Figure S11: **Sanity check of the method:** we ran the method on a Resnet50 with randomized weights, and extracted the 3 most relevant concepts for the class 'Chain saw'. When weights are randomized, concepts are mainly based on color histograms. This might result from skip connections which propagate signal from the inputs.

## H Computational cost

Although CRAFT seems like it would require a lot of resources to run, it is actually quite efficient. **Scikit-learn**'s implementation of NMF runs quite fast on the relatively small matrices we work with, and thus, a small amount of steps of ADMM are required; the computation of Sobol indices on only the last layers of the network is not very expensive; and, thanks to the efficiency of **jaxopt**, the concept-wise grad-cAM takes about as much time to calculate as the standard version (for each concept). That being said, the code in its current form doesn't support batched input images for concept-wise heatmaps, so Smoothgrad [5] and other methods based on the aggregation of gradients will take considerably longer to compute.