# OpenReview forum: "CRAFT: explaining using Concepts from Recursive Activation FacTorization"
_NeurIPS.cc/2022/Conference — NeurIPS 2022 Submitted_

### Official Review · Reviewer_B8eQ · 2022-07-08

**Rating:** 6
**Confidence:** 4
**Soundness:** 2 fair
**Presentation:** 2 fair
**Contribution:** 3 good

**Summary:**

This work presents a novel method to provide local and global explanations for the decision process of neural convolutional models. Their approach tries to divide agglomerated concepts from later layers into disentangled sub-concepts in earlier layers to provide more human-comprehensible explanations/ concepts. For this purpose, they make use of three “ingredients”, namely 1) a reiteration to find early-stage concepts that are human-comprehensible, 2) an approach from the variance-based sensitivity analysis, the Sobol indices, to estimate concept importance in the input for the prediction and 3) a method to provide an attribution map through implicit differentiation for the found concepts such that we know where the concept is contained in the input. Experiments, including some ablations studies and a user study, are provided to prove the function of the proposed method.

**Questions:**

Suppose the decision is made based on these concepts. How exactly does your method differ and compare to prototype networks? To some extent, a concept seems similar to a prototype, especially as you show multiple prototypical (parts) examples when you want to answer the “what” question.


Further remarks not affecting the evaluation:

- link to anonymous GitHub does not work
- Line 64 meanignful -> meaningful
- Line 105 vectorin -> vector in
- In general, quite some typos, capitalization, and punctuation errors
- Please incorporate fig 7 in your main text if you refer to it multiple times
- The appendix is a bit messy; many typos and references to figures are missing, etc.
- Line 245 What are KKT conditions?

**Limitations:**

No real discussion on limitations. Neither from a technical nor from a societal view. Also no discussion on future work. That’s a big downside of this work, especially if you motivate your work with societal reasons, i.e. “the right to explanations” for end-users. What impact could your method have?

**Strengths And Weaknesses:**

**Strengths**:
- S1 Using the NMF to find promising concepts is a good idea, especially to only find sparse and meaningful concepts. Using Sobol indices for feature importance seems sensible. Also, combining the three ingredients to solve this task is exciting and overall yields a novel approach to explainability. This work presents a multitude of solid mathematical solutions for practical problems in this task.
- S2 The figures used to conceptually show the function of their method is very good and intuitive.
- S3 The story is in general clear.

**Weaknesses**:
- W1 Overall, the experimental evaluation is weak. Many claims are only shown conceptually, especially comparing ACE and the failure cases. That is a big drawback. Also, the user study is very sparse. It shows that sub-concepts help to a small degree in finding an intruder. Attributing this finding to more meaningful concepts is a vast step. Also, no comparison of what other methods achieve is not given.
- W2 The work claims that the method rather answers what than where, but this question remains unanswered despite intuitively making sense. The idea is to not only highlight where a concept is attributed in the input but also what the concept looks like. However, still, it is hard to understand what the concept is. The shovel examples left me somewhat confused. I agree that this approach targets the “what” but only to a limited extent. Here, I am missing a discussion on this claim.
- W3 In general, a fair discussion on limitations like computational expense, advantages, and disadvantages, in general, are missing. How efficient is this method? How computationally expensive is this method? If it only works with ReLUs, that is a significant drawback; what could be the following steps to work with other activation functions?
- W4 The intro motivation is a bit weak. Explainability is essential, yes. However, as far as I understood, the GDPR’s “right to explanations” does not ask for an explanation this work is proposing. The GDPR is mainly about privacy rights, which means the “right to explanations” refers to the data a model is trained with, which your work does not really target. Instead, the proposed European AI act targets the “right to explanations” in your way.

Based on the exciting idea I tend to accept this paper. However, there are concerns about the evaluation. If the authors address these concerns, e.g. on the missing limitation discussion, I am happy to change my score.

---

> ### Author Response · Authors · 2022-08-02
> **Answer to Reviewer B8eq (Part I)**
>
> We are pleased that you liked our paper and found it clear.
>
> Regarding your questions:
>
> - W1) Overall, the experimental evaluation is weak. Many claims are only shown conceptually, especially comparing ACE and the failure cases. That is a big drawback. Also, the user study is very sparse. It shows that sub-concepts help to a small degree in finding an intruder. Attributing this finding to more meaningful concepts is a vast step. Also, no comparison of what other methods achieve is not given.
>
> The qualitative nature of the evaluations is a failing of concept-based XAI, as there isn’t currently a metric to mathematically measure the understandability of concepts. However, we did thoroughly evaluate CRAFT’s faithfulness/fidelity through the deletion and insertion metrics on 100000 images for 5 different seeds (for a total of half a million images) and run experiments with human subjects (73 people, with 36 ML experts and 37 laymen) where we corroborate our hypothesis of human understandability of our concepts and sub-concepts. As a basis of comparison, in ACE, they poll 30 people and run their faithfulness experiments on 1000 images. The SOTA papers on this branch of XAI mostly rely on deletion and insertion metrics, and human experiments for quantitative analysis of their results, and hence, we do so as well.
>
> Regarding our user study, we did not design the intruder test to validate whether the sub-concepts are more meaningful than the concepts, but to determine their meaningfulness as concepts themselves. To compare the meaningfulness of concepts to sub-concepts, we used the binary choice test, where we show that laymen prefer the sub-concepts 74.95% of the time, and experts, 76.1%. We revised the way this was described to improve clarity.
>
> As for the comparisons to other methods, adding another test to the ones we already do would have made the polls more complex, and given that they were voluntary submissions, we believe it would have had a negative impact on the amount of final submissions. For a comparison of concepts extracted through NMF to ACE (clustering), we refer the reviewer to [1].
>
> [1] Zhang, R., Madumal, P., Miller, T., Ehinger, K. A., & Rubinstein, B. I. (2021, May). Invertible concept-based explanations for cnn models with non-negative concept activation vectors. In Proceedings of the AAAI Conference on Artificial Intelligence (Vol. 35, No. 13, pp. 11682-11690).
>
> - W2) The work claims that the method rather answers what than where, but this question remains unanswered despite intuitively making sense. The idea is to not only highlight where a concept is attributed in the input but also what the concept looks like. However, still, it is hard to understand what the concept is. The shovel examples left me somewhat confused. I agree that this approach targets the “what” but only to a limited extent. Here, I am missing a discussion on this claim.
>
> We argue that the limitation of current attribution methods is that they only provide location, whereas CRAFT goes further by providing information about the what through the lens of concept-based explainability: "what concept does the model see here". You are right in the sense that we also suffer from the same limitation as concept-based explainability: the “what” is answered to a limited extent. We have added a discussion on the subject in the limitations section (at the beginning of the supplementary material).
>
> - W3) In general, a fair discussion on limitations like computational expense, advantages, and disadvantages, in general, are missing. How efficient is this method? How computationally expensive is this method? If it only works with ReLUs, that is a significant drawback; what could be the following steps to work with other activation functions?
>
> This is a good point. The computational cost of CRAFT is similar to ACE and the method can be run as long as a relatively large number of inferences are possible, e.g. we can generate the concepts of a class for a resnet50 in 1.000 forward passes, and hence, less than a RISE explanation for a single image.
>
> The non-negativity constraints on the activation function arise from the NMF as our choice of activation factorization, which drastically improves the quality of extracted concepts[1]. However, CRAFT can be applied to models with layers with other non-negative activations (sigmoid, softplus, exponential, etc.), and future works include working with lower-bounded activations.  This limitation has been discussed and this design choice has been kept as: NMF requires non-negativity (so any positive activation function would be compatible), which is compatible with most common architecture including Resnets, EfficientNet inceptions, SENets (non exhaustive list).
>
> In any case, we have added the information of the computation cost to the supplementary material, and a section where we talk about the limitations we’ve identified to the beginning of the appendix. We thank you for the remark.

---

> > ### Author Response · Authors · 2022-08-02
> > **Answer to Reviewer B8eq (Part II)**
> >
> > - W4) The intro motivation is a bit weak. Explainability is essential, yes. However, as far as I understood, the GDPR’s “right to explanations” does not ask for an explanation this work is proposing. The GDPR is mainly about privacy rights, which means the “right to explanations” refers to the data a model is trained with, which your work does not really target. Instead, the proposed European AI act targets the “right to explanations” in your way.
> >
> > The reviewer is right, we have corrected this statement in the main paper.
> >
> > - Q1.1) Suppose the decision is made based on these concepts. How exactly does your method differ and compare to prototype networks? To some extent, a concept seems similar to a prototype, especially as you show multiple prototypical (parts) examples when you want to answer the “what” question.
> >
> > Indeed, a concept seems similar to a prototype, but its difference resides in the fact that the former is extracted from a pre-trained network that can have any architecture – as long as its activations are non-negative in our case. For a model to generate prototypes, we are forced to train a whole new model with all that it implies (hyper-parameter tuning, additional expense due to training process, we might lose some performance due to the architecture choice, etc.). We are not saying that concept-based XAI is better in every aspect, just that it has its benefits over prototype networks (but also, its disadvantages, of course).
> >
> > - Q1.2) link to anonymous GitHub does not work
> >
> > We apologize for the mishap, it should be fully available now  (unchanged from submission on 8 May 2022).
> > NB: the full stop at the end is not part of the URL.
> >
> > - Q1.3) What are KKT conditions?
> >
> > The Karush-Kuhn-Tucker (KKT) conditions are a set of conditions fulfilled by the optimum of some convex constrained optimization problems. They can be used to characterize and find the optimum. We need them to apply the implicit differentiation theorem, since the dual variables carry the sensibility of the problem with respect to its input. The equation (10) in appendix C is explicitly built by stacking KKT conditions. We add a precision in the appendix to make this more clear.
> >
> > - L1) No real discussion on limitations. Neither from a technical nor from a societal view. Also no discussion on future work. That’s a big downside of this work, especially if you motivate your work with societal reasons, i.e. “the right to explanations” for end-users. What impact could your method have?
> >
> > In order to focus on design choices, this section was not included in the main paper. We agree with this constructive remark, and added a section accordingly at the beginning of the appendix.
> >
> > We thank the reviewer for the remarks on the form, we have revised the paper and its quality has greatly improved.
> >
> > Best regards,

---

> > > ### Comment · Reviewer_B8eQ · 2022-08-07
> > > **RE: Answer to Reviewer B8eq**
> > >
> > > Please, next time, mark the revised parts and/or describe them if you revise your manuscript. Otherwise, it takes a lot of effort to find the revised parts and check them for correctness.
> > >
> > > W3) Your limitations and broader impact statement, as well as the computational statement, read quite well, but the limitations statement must be added to the main text. This is one of the essential parts and cannot be put into the supplement. I really recommend shortening your paper in general or putting one of the technical details in the appendix instead of the limitations statement. However, I disagree with the broader impact statement: "for instance, dataset's labels are never used in this method, only the model's predictions. Without claiming to remove confirmation bias, the method focuses on what the model sees rather than what we expect the model to see." 1) the model is indeed trained with labels. So, yes, the labels are not directly used in your method, but the predictions heavily rely on the labels seen while training. 2) Hence, the model sees what the labels (i.e. the labelers' opinion) told the model to see during training.
> > >
> > > W1) Although it's indeed difficult to measure concepts' understandability, you then have to adjust your claims accordingly. Can you please add more and better understandable examples of the user study (Fig.13)?
> > >
> > > W2) Still, your claim in the end that you answer "what" doesn't fit the presented work. It is not explained why "what" is addressed especially compared to the previous explainability methods. Furthermore, I think it's at least arguable that previous methods only show "where" and not "what". Though it may be true that your method improves explaining the what, the presented way does not address this in a proper way. I again recommend working over this wording to align the story.
> > >
> > >  It's still problematic that this work claims to produce human-interpretable (meaningful) explanations. Especially the last concept in each figure (S3-S5) shows that it's very likely that people have to guess the actual concept and will come up with a variety of different ones. The authors also "complain" about user studies being very expensive (in many ways) which is true. But I don't believe there is a cheaper way around. Explainability is subjective and normative. Measuring it is best done by asking people themselves. I recommend when working on XAI to focus on precise user studies. When users prefer sub-concepts over concepts that doesn't directly imply they are more understandable or meaningful. In order to claim that, this has to be asked in the user study as well: "Please select the more meaningful conceptual explanations".

---

> > > > ### Author Response · Authors · 2022-08-08
> > > > **Re:Re: Answer to Reviewer B8eq**
> > > >
> > > >
> > > > **W3.**
> > > >
> > > > We agree and upon acceptance, the extra page will be dedicated to the broader impact and limitations sections.
> > > >
> > > > Regarding the use of labels, we wish to avoid user confirmation bias - i.e. the user has unconscious expectations of what the explanation should look like for a given class before it is generated. To avoid this, the fact that we use model predictions instead of labels allows us to capture concepts that are important to the model, despite not having the correct predictions (e.g. ski suits and dirt are not necessarily part of shovel images but may come from images that the model predicts as shovels).
> > > >
> > > > **W1.**
> > > >
> > > > We have added some additional examples of the user study to the appendix.
> > > >
> > > > **W2.**
> > > >
> > > > > I think it's at least arguable that previous methods only show "where" and not "what".
> > > >
> > > > This is indeed a criticism that is often made to attribution methods and this is the reason why we developed CRAFT: to go beyond the where. Concerning the "what", it is true that we appeal to users’ (arguably innate) capability of finding patterns in groups of images to give a name to each concept, it still constitutes an important step forward in explaining the "what" when compared to what’s currently available in the literature of post-hoc XAI.
> > > >
> > > > We have added a clarification regarding this in the limitation section.
> > > >
> > > >
> > > > We claim to generate human interpretable concepts and for this purpose we performed two controlled psychophysics experiments with humans (more than 70 people), from different audiences (experts, laymen).
> > > > In addition, non-understandable concepts would have been unable to achieve scores as high as those in the intruder experiment.
> > > > Contrary to what is said, we did not ask the participants to choose between sub-concept and concept (we agree that this would not be sufficient) but we set up the experiment of asking humans which cluster was most likely to contain an image - that image itself belonging to the cluster and a subcluster. The results show that humans find the sub-clusters more coherent / likely to contain the image.
> > > >
> > > > We have clarified this point in the experimentation section
> > > >
> > > > Thank you for this constructive discussion. We believe that, as a result, these clarifications will significantly improve the paper.

---

### Official Review · Reviewer_tkuX · 2022-07-09

**Rating:** 4
**Confidence:** 4
**Soundness:** 2 fair
**Presentation:** 2 fair
**Contribution:** 3 good

**Summary:**

In this paper the authors propose the “Concept Recursive Activation FacTorization” (CRAFT), a framework to generate concept-based explanations from trained networks. CRAFT aims at addressing current limitations in state-of-the-art concept-based techniques (e.g. ACE [1]) including: (i) the association of *one concept only* for each image segment/patch, (ii) the instability/noise of gradients w.r.t. concept vectors, (iii) the localization of concepts in the input space to provide local explanations. To address these limitations, the authors equip CRAFT with three methods: (i) a recursive procedure to automatically decompose concepts into sub-concepts, (ii) an alternative scoring function for concept importance based on Sobol indeces, and (iii) a method to localize concepts in the input (pixel) space. Following Zhang et al. [2], CRAFT identifies coherent bases of concepts using the Non-negative Matrix Factorization (NMF) rather than k-Means as in ACE [1]. In the experiments the authors show that CRAFT can: (i) identify meaningful and coherent concepts, (ii) localize concepts in the input space allowing users to explain misclassified images, (iii) decompose high-level concepts into coherent subconcepts, and (iv) provide a more robust ranking concept w.r.t. TCAV in terms of  fidelity scores.

[1] Ghorbani, Amirata, et al. "Towards automatic concept-based explanations." Advances in Neural Information Processing Systems 32 (2019).

[2] Zhang, Ruihan, et al. "Invertible concept-based explanations for cnn models with non-negative concept activation vectors." Proceedings of the AAAI Conference on Artificial Intelligence. Vol. 35. No. 13. 2021.

**Questions:**

Addressing the main limitations mentioned above would significantly improve the paper and, as a consequence, reviewers’ feedback and scores. However, I understand that the authors may not have enough time to address them all in the rebuttal. For this reason I list here a few minor items which can give the authors a better chance to improve their work in the short term:
1. A few sentences may need further clarification as they may contain typos or have grammar issues (e.g. ““The second concept elucidated by craft is located on the astronaut’s pants that **he confuse** with the ski suits” (Figure 5)). The authors may also improve the flow using the active voice more frequently (e.g. “Namely, in [38] and [39], explanations are generated as the inputs” (line 97)). Other sentences which may benefit from rephrasing include: “Once the factorization done, and the reconstruction of the activation of the image can then be interpreted as a combination of a set of concepts and a coefficient associated to these concepts.” (line 116), “A common concern with concept extraction methods is that make sense to humans does not mean it” (line 189), “Finally, we can capture the importance that a concept might have as a main effect – by itself – as well 210 as by interacting with other concepts on the model’s output by calculating the expected variance that 211 would remain if all the indices of the masks except the Mi were to be fixed.” (line 209).
2. A few acronyms need to be spelled out (e.g. TCAV, line 112).
3. The authors reference Figure 3 before Figure 2. I would suggest to swap the figures.
4. Some claims may require to be rephrased as in their current form may not align with actual results. For instance, I find the possibility of using CRAFT for explaining complex failure cases quite exciting, but the only experimental result is on one cherry-picked example which may not be enough to prove the efficacy of the method (Section 4.2).  Another example is when authors state “We use our approach and combine local and global explanation, to *accurately* explain predictions” (line 72). Here I was expecting a *quantitative* result in the experimental section showing the actual *accuracy* scores of CRAFT’s explanations. A third example is when authors state that “CRAFT yields more interpretable concepts than ACE” (line 74). The main issue here is that ACE and CRAFT are different under many aspects. These differences make a fair comparison a bit tricky as each difference may introduce a confounding factor in the comparison. However, the results here are only qualitative (Figure 4 and appendix) which makes the claim a bit weaker than it could have been. Moreover, the experiment design does not allow to identify which ingredient (i.e. Sobol indeces? NMF? or $\tau$?) from ACE or CRAFT is the one that is actually making the difference. A fourth example is when authors state “These examples illustrate one of the weaknesses of ACE: the segmentation used can introduce biases through the baseline value used. Moreover, this segmentation does not facilitate the interpretation of the concepts” (line 276). However the evidence here is only based on a qualitative visual comparison which may not be enough to support such strong claims.

**Limitations:**

As mentioned before, the authors may consider to include a description for the limitations of their work. Including relevant limitations would significantly improve the paper as it would allow other scientists to understand how to use and improve CRAFT advancing the research field. A possible limitation worth mentioning is that the experimental evidence is mostly qualitative (Section 4.1, 4.2, and 4.3) and/or possibly subject to confirmation or selection biases (Section 4.3).

**Strengths And Weaknesses:**

#  Strengths
The automatic decomposition of high-level concepts into sub-concepts is the most interesting contribution of this paper w.r.t. existing concept-based approaches. This is a significant step forward w.r.t. state-of-the-art approaches such as ACE as it enables the identification of coherent hierarchies of concepts. In particular concept hierarchies may facilitate human experts in navigating the concept space. Even though the methods are not new (e.g. NMF, or Sobol indeces), they are effectively used in new contexts. In addition, the experiments show how CRAFT can be used in practice as a powerful tool to “debug” misclassified samples by localize concepts in the input space.

# Limitations
While this paper introduces a few interesting steps forward, the authors may consider a few suggestions for improving their contributions. In particular the authors may consider to:
1. Broaden the experimental section as it is currently based on one dataset only, thus questioning the generality of the results. I would strongly encourage the authors to broaden their experiments to include additional benchmarks commonly used in concept-based XAI papers (e.g. CUB [1], TabulaMuris [2], or CelebA [3]).
2. Include a description for the limitations of their work. Including relevant limitations would significantly improve the paper as it would allow other scientists to understand how to use and improve CRAFT advancing the research field. A possible limitation worth mentioning is that the experimental evidence is mostly qualitative (Section 4.1, 4.2, and 4.3) and/or possibly subject to confirmation or selection biases (Section 4.3).
3. Include critical information which is currently missing for understanding and reproducing the experiments. For instance, the authors do not mention what error bars represent (e.g. Figure 6) nor how the images where selected for the human experiments i.e., they just mention that “the choice was randomized” (line 701), but they do not specify how (e.g. random selection from cherry-picked examples or from all images? From groups of image crops or from the original images?). Another issue is that it is not clear how the authors pick CRAFT hyperparameters (e.g. $\tau$, line 263) and how sensitive the results are to hyperparameters’ changes.
4. Reorganize some sections, include missing information, or rephrase a few sentences to improve flow and clarity. For instance, in the abstract the authors do not mention any knowledge gap which makes a bit unclear why CRAFT might be needed. On the contrary, in the introduction the authors mention four knowledge gaps (lines 31, 41, 47, and 51) which makes a bit unclear what is the *main* purpose of CRAFT. Making the introduction and contributions more focused would significantly improve flow and clarity and it would save the authors some space for describing methods, results, and limitations.

[1] Wah, Catherine, et al. "The caltech-ucsd birds-200-2011 dataset." (2011).

[2] Cao, Kaidi, Maria Brbic, and Jure Leskovec. "Concept learners for few-shot learning." arXiv preprint arXiv:2007.07375 (2020).

[3] Liu, Ziwei, et al. "Large-scale celebfaces attributes (celeba) dataset." Retrieved August 15.2018 (2018): 11.

---

> ### Author Response · Authors · 2022-08-02
> **Answer to Reviewer tkuX (Part I)**
>
> Thank you for reading our paper and taking an interest in it. We will address your concerns separately.
>
> - L1) Broaden the experimental section as it is currently based on one dataset only, thus questioning the generality of the results. I would strongly encourage the authors to broaden their experiments to include additional benchmarks commonly used in concept-based XAI papers (e.g. CUB [1], TabulaMuris [2], or CelebA [3]).
>
> We have adopted the benchmarks and datasets of the papers doing automatic concept discovery [1,2], which focus mainly on ImageNet.
>
> [1] Ghorbani, A., Wexler, J., Zou, J. Y., & Kim, B. (2019). Towards automatic concept-based explanations. Advances in Neural Information Processing Systems, 32.
> [2] Zhang, R., Madumal, P., Miller, T., Ehinger, K. A., & Rubinstein, B. I. (2021, May). Invertible concept-based explanations for cnn models with non-negative concept activation vectors. In Proceedings of the AAAI Conference on Artificial Intelligence (Vol. 35, No. 13, pp. 11682-11690).
>
> - L2.1) Include a description for the limitations of their work. Including relevant limitations would significantly improve the paper as it would allow other scientists to understand how to use and improve CRAFT advancing the research field.
>
> We have added a section on limitations and broader impact in the supplementary material which focuses on what we believe are ways to improve the method and how we hope it will impact the field of XAI.
>
> - L2.2) A possible limitation worth mentioning is that the experimental evidence is mostly qualitative (Section 4.1, 4.2, and 4.3) and/or possibly subject to confirmation or selection biases (Section 4.3).
>
> We can hardly say that our evidence is mostly qualitative or that we are subject to confirmation or selection biases. Firstly, we have measured the faithfulness of the extracted concepts on more than 100000 images on 5 different seeds (so, half a million in total), thus verifying that concepts are indeed important to the model for predicting correctly and making sure that we are not suffering from confirmation bias. Secondly, we measure the understandability of our concepts through our human experiments on 73 people coming from different environments (ML experts and laymen), all pointing to the clarity of our extracted concepts, and the benefit of explaining using sub-concepts. This kind of experiment is quite time consuming to set up and analyze, and it’s not done in a lot of papers of the field. In ACE[1], they perform the same experiments (with 30 people vs 73, and they test deletion and insertion on 1000 images, instead of half a million). As for the other results, they are easy to reproduce with the code that we uploaded to the anonymous repository.
>
> We do agree that the lack of a metric for mathematically comparing the understandability of concepts is a failing of the concept-based XAI field, and we will  address it in future work.
>
> [1] Ghorbani, A., Wexler, J., Zou, J. Y., & Kim, B. (2019). Towards automatic concept-based explanations. Advances in Neural Information Processing Systems, 32.
>
> - L3) Include critical information which is currently missing for understanding and reproducing the experiments. For instance, the authors do not mention what error bars represent (e.g. Figure 6) nor how the images where selected for the human experiments i.e., they just mention that “the choice was randomized” (line 701), but they do not specify how (e.g. random selection from cherry-picked examples or from all images? From groups of image crops or from the original images?). Another issue is that it is not clear how the authors pick CRAFT hyperparameters (e.g. , line 263) and how sensitive the results are to hyperparameters’ changes.
>
> The reviewer is right. We have added more details in section “Fidelity experiments” of the supplementary material. Error bars are obtained using multiple seeds (5) on 100,000 images each time and represent the standard deviation. Regarding the way the images were chosen for the human experiment: once the NMF is trained on random ImageNet image crops (for a given class), the crops with the highest $U_i$ (thus the most representative of the concept) were automatically chosen.
>
> As to the choice of hyperparameters, we chose $r = 25$ to compare to ACE (which did so as well), but didn’t notice much difference when slightly changing this parameter, and the same can be said about the crop sizes. We randomly picked crops of $64 \times 64$ pixels, but altering the size to, for instance, $50 \times 50$ doesn’t have any real impact. We’re of course leaving out hyperparameter choices that would never work (e.g. crops of $5 \times 5$ pixels, $r = 2$, etc.). Optimally choosing $r$ as the non-negative rank of the activations matrix is an NP-hard problem, and we leave some improvements on how we perform the factorization for future works.

---

> > ### Author Response · Authors · 2022-08-02
> > **Answer to Reviewer tkuX (Part II)**
> >
> > - L4) Reorganize some sections, include missing information, or rephrase a few sentences to improve flow and clarity. For instance, in the abstract the authors do not mention any knowledge gap which makes a bit unclear why CRAFT might be needed. On the contrary, in the introduction the authors mention four knowledge gaps (lines 31, 41, 47, and 51) which makes a bit unclear what is the main purpose of CRAFT. Making the introduction and contributions more focused would significantly improve flow and clarity and it would save the authors some space for describing methods, results, and limitations.
> >
> > Thank you for your suggestions, we have taken them into account, and the resulting document is much improved.
> >
> > Concerning your first 3 questions that we should improve during the rebuttal, thank you for pointing this out to us, we have corrected the typos and reworded the corresponding sentences.
> >
> > - Q4.1) Some claims may require to be rephrased as in their current form may not align with actual results. For instance, I find the possibility of using CRAFT for explaining complex failure cases quite exciting, but the only experimental result is on one cherry-picked example which may not be enough to prove the efficacy of the method (Section 4.2).
> >
> > The last experiment of the paper proposes to tackle the problem of understanding complex failure cases in XAI which the current methods do not address. It is true that we have only shown some examples in the paper, and we are working to add some more to the supplementary material, but we have made the code available, which is easy to use so that everyone can test it.
> >
> > It is also true that there is no benchmark for this sort of application at the moment. Nevertheless, the proposal of such a dataset is a separate work that we have not addressed in this paper, but that it’s much needed in the field of concept-based XAI.
> >
> > - Q4.2) A third example is when authors state that “CRAFT yields more interpretable concepts than ACE” (line 74). The main issue here is that ACE and CRAFT are different under many aspects. These differences make a fair comparison a bit tricky as each difference may introduce a confounding factor in the comparison. However, the results here are only qualitative (Figure 4 and appendix) which makes the claim a bit weaker than it could have been. Moreover, the experiment design does not allow to identify which ingredient (i.e. Sobol indeces? NMF? Or $\tau$ ?) from ACE or CRAFT is the one that is actually making the difference.
> >
> > We have not designed CRAFT as an iterative improvement over ACE, but as a new method as a whole, capable of generating both global and local explanations based on concepts inside the NN that are identified without supervision. Upon reading the original ACE paper, we remark that every design choice was made to furnish 25 databases of “concept images” to then leverage TCAV.  Instead, we exploit the compositional nature of images by inputting crops into the network and then searching for these concepts inside the network without supervision and with a method that allows us to identify its provenance on the input later. We leverage the results of the experiments on human subjects in [1] to say that concepts generated through NMF are more meaningful to humans than those extracted using ACE, and the deletion and insertion metrics to demonstrate that our concept importance estimation (Sobol indices) is better than TCAV.
> >
> > [1] Zhang, R., Madumal, P., Miller, T., Ehinger, K. A., & Rubinstein, B. I. (2021, May). Invertible concept-based explanations for cnn models with non-negative concept activation vectors. In Proceedings of the AAAI Conference on Artificial Intelligence (Vol. 35, No. 13, pp. 11682-11690).
> >
> > - Q4.3) A fourth example is when authors state “These examples illustrate one of the weaknesses of ACE: the segmentation used can introduce biases through the baseline value used. Moreover, this segmentation does not facilitate the interpretation of the concepts” (line 276). However the evidence here is only based on a qualitative visual comparison which may not be enough to support such strong claims.
> >
> > With regard to segmentation+baseline introducing a bias, we have added citations to back up the claim that the baseline value is a problem (which is a known problem in the XAI community). Regarding the second point, the reviewer is right and we have removed this sentence.
> >
> >  Best regards,

---

> > ### Comment · Reviewer_tkuX · 2022-08-03
> > **RE: Answer to Reviewer tkuX**
> >
> > I thank the reviewers for their effort. I'm happy they found some comments useful. I'm raising my score a bit as I really appreciated your work and ideas. **However, I still think the current manuscript is not ready for being accepted, yet.** The main reason is that the experimental section is too limited in terms of datasets and quantitative evaluations, which is a common concern I share with other reviewers (Rev-p9rf and Rev-B8eq).
> >
> > Please consider the following are just my personal opinions and I'm more than happy to be convinced otherwise.
> >
> > > We have adopted the benchmarks and datasets of the papers doing automatic concept discovery [1,2], which focus mainly on ImageNet.
> >
> > I know the papers and understand the authors point of view. However, your argument is not sound i.e., it is an example of the "appeal to authority" logical fallacy. The reasons why I suggested to broaden your experimental section are:
> > - it's really hard to argue that a method A works based on the results on 1 dataset only as there might be intrinsic biases in the given dataset—exploited by the method A—that do not hold on other datasets
> > - limited experimental sections (as in ACE and other concept papers) are actually weakening this field
> >
> > I personally value novelty more than SOTA results and excessive experiments. However, assessing novelty is part of the game and for top tier venues my personal expectation is an assessment on more than 1 dataset.
> >
> >
> > > We can hardly say that our evidence is mostly qualitative
> >
> > I understand that the volume of your tests is huge and your human experiments is extensive. However, as you pointed out, my main concern is the lack of a metric for mathematically measure the quality of your work. Testing on a lot of samples is good, but **without a mathematically grounded metric and diverse datasets is quite difficult to draw general and convincing conclusions**. Let me be clear: I liked your idea, but your claims are quite weak without a assessment thorugh a quantitative and mathematically grounded metric. Concerning the human experiments, I agree that your analysis of the outcomes is quantitative, but my concern is that the human evaluations are still subjective and qualitative: they are extremely useful for user testing (and related claims), but they cannot replace a mathematically grounded metric.

---

> > > ### Author Response · Authors · 2022-08-05
> > > **RE:RE: Answer to Reviewer tkuX**
> > >
> > > Thanks for your prompt reply and for raising our score, and we’re glad you appreciated our work and ideas. We understand your concerns, and quite frankly, we do agree that concept-based XAI as a whole lacks mathematically-grounded metrics for human understandability, and the dataset diversity is not as large as it should be. In fact, attribution-based explainability started out the same way: with most evaluations being qualitative and severe confirmation bias. With time, the need for a way to compare the different methods steered researchers to propose metrics for the different desiderata – e.g. insertion and deletion for faithfulness. We do hope that if our method gets some traction and gets picked up by other research teams, this need for a metric will push more people into this topic and eventually a solution will be found.
> > >
> > > As for the lack of diversity of datasets, we simply chose to focus on more intensively testing on the dataset that other concept-based methods used, as showing results in one on which no other method was benchmarked wouldn’t allow for a fair comparison. Of course, this was a design choice whose optimality is debatable. We do intend to test it on other datasets to verify its validity, but we are confident that if it works on natural images, it should work in other (arguably easier) contexts
> > >
> > > We sincerely thank you for this great discussion!

---

### Official Review · Reviewer_KNMa · 2022-07-18

**Rating:** 5
**Confidence:** 3
**Soundness:** 3 good
**Presentation:** 2 fair
**Contribution:** 2 fair

**Summary:**

The authors have done a nice job by improving the concept based explanation generation techniques from different perspectives. An automatic concept extraction followed by a recursive procedure towards achieving a set of concepts and sub-concepts are proposed in this work. A concept-based heatmap generation method is also proposed here that backpropagate concept values to pixel space that is very similar to widely used class-based saliency methods.

**Questions:**

L31: What do you mean by information methods?

L73-74: If some figure in the supplementary section is referred to in the main paper, it is a good practice to clearly mention that.

L137: I request the authors to explain the notation h_{l,k}, as l and k represent some layer and the total number of layers respectively, but it was not clear to me how they're used in a single notation.

L163: Please write the full form when the short form is used for the first time in the paper. The full form of NNLS (i.e. Non Negative Least Square) should have been mentioned here.

Other questions:
a) Please explain the notations of Table 1 in detail. It is not very clear what 'n' represents here.

b) Here the concepts found by NMF are claimed to be human understandable. If any or some of the concepts are exceptions, I am interested to know whether these concepts are specific to a particular category?

c) As the gradient based explanation generation techniques generally produce noisy saliency maps, did you see any such phenomenon for concepts? Do the heatmaps for all the images are as nice as shown in fig.1? If yes, which part of the method (related to concept attribution map generation) helped you to achieve that specifically?

**Ethics Review Area:**

["I don’t know"]

**Limitations:**

The authors did not add any section to address the limitations and potential negative societal impact of this work.

**Strengths And Weaknesses:**

Strengths:
a) Several important directions are worked on here, that start with concept generation in a recursive manner from multiple layers of the network. This technique helps the cases where concepts generated from a very deep layer don't make much sense and decomposing them with sub-concepts can develop an improved understanding of the model.

b) This is the first method, I believe, to work on saliency based explanations for each concept. The authors have shown the effectiveness of this approach by explaining different concepts in a single image.


Weaknesses:
a) Although the quality of a technical literature shouldn't be judged by the language quality, I still believe that there are many scopes to polish the language and take care of some typos to improve the readability of the paper.

b) I couldn't access the code related to the paper from the provided link. If by any chance the code was not ready for sharing by the submission time, it would have been better to skip that.

---

> ### Author Response · Authors · 2022-08-02
> **Answer to Reviewer KNMa**
>
> Thank you for taking an interest in our paper and for the interesting questions.
>
> We will answer the questions sequentially.
>
> - Wa) Although the quality of a technical literature shouldn't be judged by the language quality, I still believe that there are many scopes to polish the language and take care of some typos to improve the readability of the paper.
>
> We recognize that some typos slipped through to the final submission, that readability could have been improved, and we apologize for it. We have revised it and its quality has much improved.
>
> - Wb) I couldn't access the code related to the paper from the provided link. If by any chance the code was not ready for sharing by the submission time, it would have been better to skip that.
>
> We apologize for the mishap, it should be fully available now  (unchanged from submission on 8 May 2022).
> NB: the full stop at the end is not part of the URL.
>
>
> - L31: What do you mean by information methods?
>
> Apologies, this has been fixed. We meant “Attribution methods”.
>
> - L73-74: If some figure in the supplementary section is referred to in the main paper, it is a good practice to clearly mention that. L137: I request the authors to explain the notation h_{l,k}, as l and k represent some layer and the total number of layers respectively, but it was not clear to me how they're used in a single notation. L163: Please write the full form when the short form is used for the first time in the paper. The full form of NNLS (i.e. Non Negative Least Square) should have been mentioned here.
>
> Thank you. We relabeled figures in the supplementary information. Regarding $h_{l,k}$, it denotes the top-most part of the neural network, going from layer $l$ to the output/logits.
>
> - Other questions: a) Please explain the notations of Table 1 in detail. It is not very clear what 'n' represents here.
>
> For the human psychophysical experiments, we separated our poll’s results into two groups: ML experts (for a total of 36 people), and laymen (37 people) – the $n$ representing the amount of people in each group. Then, we divided the table in two, one part for each task (intruder detection and binary choice between the concept and the sub-concept). In essence, our results prove (with more or less statistical significance) that users prefer subconcepts over concepts for understandability (binary choice test), and that they are capable of identifying the intruder when clusters are generated using our NMF factorization method (intruder test).
>
> - b) Here the concepts found by NMF are claimed to be human understandable. If any or some of the concepts are exceptions, I am interested to know whether these concepts are specific to a particular category?
>
> This is a great question, which we plan to investigate in future work. Meanwhile, we refer the reviewer to the supplementary information where we show all the top most important concepts for each class tested (which are not cherry-picked but are the classes in the ImageNette subset), and seem to be understandable by humans according to our human experiments.
>
> - c) As the gradient based explanation generation techniques generally produce noisy saliency maps, did you see any such phenomenon for concepts? Do the heatmaps for all the images are as nice as shown in fig.1? If yes, which part of the method (related to concept attribution map generation) helped you to achieve that specifically?
>
> Thank you for this excellent question. As a matter of fact, we did obtain noisy explanations at first when we got saliency maps to work. However, we then switched to GradCAM and started getting the smooth explanations that we showcase in the paper. It would be interesting to see if this problem persists for other attribution methods when the chosen layer is an early layer in the NN.
> We have also added a limitations and broader impact section to the beginning of the supplementary material.
>
> Best regards,

---

### Official Review · Reviewer_p9rf · 2022-07-19

**Rating:** 6
**Confidence:** 3
**Soundness:** 3 good
**Presentation:** 3 good
**Contribution:** 3 good

**Summary:**

This paper proposed a concept-based explainability method, CRAFT, to generate both global and local concept-based explanations.
This paper introduces some key contributions: (1) recursively decomposing higher-level concepts into more elementary ones.
(2) concept importance estimation based on Sobol's indices. (3) Concept Attribution Maps, which connect global concepts to local image level explanation.

**Questions:**

(1) How could we use the explanation to explain the error cases made by the model?

(2) It would be good to compare with a related paper: "A Peek Into the Reasoning of Neural Networks: Interpreting with Structural Visual Concepts" which also provides both global and local concept-level explanations.

**Limitations:**

More quantitative experiments to show the performance and compare it with other baselines would be better.


**Strengths And Weaknesses:**

**Strengths**

(1) Concept-level explanation is a challenge and an important direction. The proposed method is useful and novel, especially for the recursive decomposition to get different level concepts and the linking between global high-level concepts and local attribution map.

(2) The three ingredients based on the concept activation factorization are promising and inspiring.

(3) The paper is well organized and easy to follow.

**Weaknesses**

(1) How to verify the faithfulness of the discovered concepts? What if the human-understandable concepts are not the correct reasoning logic for the original neural networks.

(2) How to evaluate the scalability to larger classes?

(3) Are there quantitative results to show the performance and compare it with other baselines?

==========================Pose Rebuttal=============================

Thanks for your feedback. I will keep my original score.

---

> ### Author Response · Authors · 2022-08-02
> **Answer to Reviewer p9rf**
>
> We are pleased that you found our paper interesting and liked the way it’s organized.
>
> We will address each question separately.
>
> - W1.1) How to verify the faithfulness of the discovered concepts?
>
> We report improvements using the two main faithfulness measures used in the field: Deletion and insertion. A lower deletion indicates better fidelity, as does a higher insertion. As shown in Fig.6 and Fig.14, we improve the faithfulness of the discovered concepts by using Sobol indices with respect to those extracted with TCAV importance.
>
> - W1.2) What if the human-understandable concepts are not the correct reasoning logic for the original neural networks.
>
> We make sure that concepts follow the model’s logic by testing their faithfulness through the Deletion and Insertion scores. If the concepts for a given class were not to explain the model, it would instantly reflect negatively in these metrics.
>
> - W2) How to evaluate the scalability to larger classes?
>
> We assume the reviewer refers to the size of the images. The proposed method does not require more forward passes than SOTA explainability methods such as RISE hence scalability is not really an issue here. Furthermore, our experiments on ImageNet were performed on a single GPU (V100) for a ResNet50.
>
> - W3) Are there quantitative results to show the performance and compare it with other baselines?
>
> For the Sobol ingredient, we refer the reviewer to Fig.6 and Fig.14 demonstrating that the proposed approach yields a much better fidelity score than SOTA method TCAV (ACE) irrespective of the factorisation method used (PCA, NMF, RCA, ICA). For the quality of the extracted concepts, we leverage the extensive human experiments in [1] that show the improvements in terms of human understandability over clustering (as done in ACE), and our own to show the preference of users for sub-concepts over concepts. Unfortunately, there is no existing mathematical metric to measure this objectively, and thus, to compare quantitatively to other concept extraction techniques directly. We added a discussion on this topic on the limitations section (at the beginning of the appendix).
>  [1] Zhang, R., Madumal, P., Miller, T., Ehinger, K. A., & Rubinstein, B. I. (2021, May). Invertible concept-based explanations for cnn models with non-negative concept activation vectors. In Proceedings of the AAAI Conference on Artificial Intelligence (Vol. 35, No. 13, pp. 11682-11690).
>
> - Q1) How could we use the explanation to explain the error cases made by the model?
>
> We refer the reviewer to section 4.2. We are working to add more examples of this use-case to the supplementary material.
>
> - Q2) It would be good to compare with a related paper: "A Peek Into the Reasoning of Neural Networks: Interpreting with Structural Visual Concepts" which also provides both global and local concept-level explanations.
>
> We weren’t familiar with this paper at the time of writing, but we found it quite interesting, especially its capacity to generate contrastive explanations (‘why not’). However, their method proposes to learn a model on top of the extracted concepts, while ours integrates everything seamlessly into the concept extraction block – i.e. building a graph to link concept vs backpropagating through the concept extraction to locate them. We think they might work in complement, for instance by extracting concepts using the NMF and then learning the graph with them instead of ACE, thus having the best of both worlds.
>
> Best regards,

---

### Meta-Review · Area_Chair_UGg3 · 2022-08-25

**Recommendation:** Reject
**Confidence:** Less certain

**Metareview:**

Reviewers generally agreed that this paper is innovative (the decomposition of high-level concepts into sub-concepts in particular sets this paper apart from existing concept-based methods), and appreciated its potential practical utility for the explainability community (for example by providing localization in input space in addition to concepts, which can be used to debug model errors).
All Reviewers however also agreed on the main weaknesses, i.e. the limits of the validation of the method (which is restricted to only 1 dataset and lacking rigorous quantitative metrics), and a related lack of clarity in terms of technical motivation and use cases enabled for end-users of the method.
Despite some useful clarifications and improvements in the presentation of results provided in the rebuttals and the ensuing exchange, Reviewers were left unsatisfied by the responses addressing the mentioned main weaknesses, which raised renewed concerns about their seriousness.
As a result, the discussion after the rebuttal period was marked by the opinion expressed by two Reviewers that the paper's weaknesses outweigh the (albeit clear) merits and that in its current version the paper is not ready for being accepted.

**Award:**

No

---

### Decision · Program_Chairs · 2022-09-14

Reject